# NMR Spectroscopy Applied to the Metabolic Analysis of Natural Extracts of *Cannabis sativa*

**DOI:** 10.3390/molecules27113509

**Published:** 2022-05-30

**Authors:** Maria Francesca Colella, Rosachiara Antonia Salvino, Martina Gaglianò, Federica Litrenta, Cesare Oliviero Rossi, Adolfo Le Pera, Giuseppina De Luca

**Affiliations:** 1Department of Chemistry and Chemical Technologies (CTC), University of Calabria—UNICAL, Via P. Bucci 14C, 87036 Arcavacata di Rende, Italy; mariafrancesca.colella@unical.it (M.F.C.); rosachiara.salvino@unical.it (R.A.S.); martina.gagliano@unical.it (M.G.); cesare.oliviero@unical.it (C.O.R.); 2Department of Biomedical, Dental and Morphological and Functional Imaging Sciences (Biomorf), University of Messina, Polo Universitario dell’Annunziata, 98168 Messina, Italy; federica.litrenta@unime.it; 3Calabra Maceri e Servizi s.p.a., Via M. Polo 54, 87036 Rende, Italy; laboratorio@calabramaceri.it

**Keywords:** *Cannabis sativa* L., cannabinoid extraction, NMR spectroscopy, qNMR, metabolic profile, principal component analysis

## Abstract

*Cannabis sativa* is a herbaceous multiple-use species commonly employed to produce fiber, oil, and medicine. It is now becoming popular for the high nutritional properties of its seed oil and for the pharmacological activity of its cannabinoid fraction in inflorescences. The present study aims to apply nuclear magnetic resonance (NMR) spectroscopy to provide useful qualitative and quantitative information on the chemical composition of seed and flower *Cannabis* extracts obtained by ultra-sound-assisted extraction, and to evaluate NMR as an alternative to the official procedure for the quantification of cannabinoids. The estimation of the optimal ω-6/ω-3 ratio from the ^1^H NMR spectrum for the seed extracts of the *Futura 75* variety and the quantitative results from the ^1^H and ^13^C NMR spectra for the inflorescence extracts of the *Tiborszallasi* and *Kompolti* varieties demonstrate that NMR technology represents a good alternative to classical chromatography, supplying sufficiently precise, sensitive, rapid, and informative data without any sample pre-treatment. In addition, different extraction procedures were tested and evaluated to compare the elaboration of spectral data with the principal component analysis (PCA) statistical method and the quantitative NMR results: the extracts obtained with higher polarity solvents (acetone or ethanol) were poor in psychotropic agents (THC < LOD) but had an appreciable percentage of both cannabinoids and triacylgliceroles (TAGs). These bioactive-rich extracts could be used in the food and pharmaceutical industries, opening new pathways for the production of functional foods and supplements.

## 1. Introduction

*Cannabis sativa* is a fast-growing annual dioecious weed, probably native to Central Asia and the Indian subcontinent [1,2], belonging to the *Cannabaceae* family (order *Urticales*) [3]. Despite its critical taxonomic definition, because of its complex chemical composition and the presence of several spontaneous generations of hybrid species, nowadays, classifying *Cannabis* as a highly polymorphic and hybridized monotypic genus (*Cannabis sativa* L.) is the most accepted definition. Cannabis is one of the oldest and most versatile sources of intoxicating resin, textile fiber, and phytocannabinoids, which are extracted from different parts of the plant, especially from the inflorescence and for seed oil. Hemp seed oil, obtained from *Cannabis sativa L.* seeds, is highly appreciated for its nutritional, anti-inflammatory, antioxidant, and immune-stimulating properties [4]. It is practically free of cannabinoids [5], so it has no psychoactive action but, like other common vegetable oils, it is rich in essential fatty acids [6]. As reported in several works, this oil is a rich source of ω-3 and ω-6 polyunsaturated fatty acids (almost 80%), in particular, linoleic acid (LA) and α-linolenic acid (αLA), with a ω-6/ω-3 ratio approximately equal to 3:1 [7]. Although various factors, such as cultivation area, cultivar, seed origin, agronomic cultivation practices, etc., affect both the chemical composition and the ω-6/ω-3 ratio [4,7], this ratio is considered an optimal nutritional value in the prevention of the risk of coronary heart disease [8,9]. Due to this characteristic, cannabis seed oils are authorized and widely used in the food sector [10], such as in the production of functional foods. Despite the growing interest in this product, specific regulations to evaluate its analytical quality parameters are still lacking [7]. In this context, it would be desirable to find methodologies that can provide useful and rapid information both on the chemical composition and on the important ω-6/ω-3 ratio.

The female inflorescences of the *Cannabis* plant have been widely used in the traditional medicine of different populations thanks to the pharmacological activity of some phytocannabinoids present in large quantities in these parts of the plant [11]. Phytocannabinoids are a class of terpenophenolic compounds with a 21-carbon backbone: 120 of these molecules naturally present in the plant have been identified and isolated to date [3,12,13]. These natural molecules are different from synthetic cannabinoids, generally used as therapeutic agents, and from endocannabinoids, which are endogenous lipid-based retrograde neurotransmitters capable of interacting with cannabinoid receptors in the human body [14]. The renewed and recent interest in cannabis is due to the identification of these molecules, whose different pharmacological activities such as anti-inflammatory action, cell growth inhibition, and tumor regression seem to be supported by numerous experimental evidence [15,16,17]. The chemical structures of the most common cannabinoids present in the cannabis plant are shown in Figure 1. Among them, the most representative are the well-known psychotropic agent Δ^9^-trans-tetrahydrocannabinol (Δ^9^-THC), cannabidiol (CBD), and its precursor cannabidiolic acid (CBDA). Compared to THC, CBD shows therapeutic benefits without euphoric or dysphoric effects, which is an advantage for clinical applications [15,16,17]. CBD has become very popular over the years for its health benefits, and nowadays is commercially available as a dietary supplement, a lotion, and most importantly, as a CBD oil. Indeed, the interest from the scientific community in the therapeutic potential of CBD oil is growing every day [18]. The reason is simple: it has already been used in various scientific studies for the treatment of numerous health problems, and is now recognized as one of the main elements of the so-called “therapeutic *Cannabis*” [19,20,21]. Industrial hemp crops, with a low THC content, have always been exploited as food and as a source of textile fibers, but they disappeared in the 1970s due to their association with the type of plants rich in THC [22]. The reintroduction of the cultivation of some hemp cultivars to produce fibers and seeds with a THC content lower than 0.2% *w*/*w* took place only several years later, i.e., in 2009, by means of an appropriate regulation published by the European Union [23]. Nowadays, in many countries, *Cannabis sativa* cultivations and medicines have been legalized under certain conditions due to their immense prospects in various medicinal applications [24,25]. The Italian legislation on *C. sativa* cultivation is somewhat ambiguous regarding the legal and illegal uses and cultivation of the plant, and differs based on the concentration of psychoactive cannabinoids. The law 242/2016 “Dispositions for the promotion of cultivation and supply chain of agro-industrial hemp” [26] is the most recent regulation in that direction, and is the reference text governing industrial hemp production in Italy for fiber or other industrial uses different from pharmaceutics, with cultivation based around certified seeds [27]. This measure establishes that the THC level must not exceed 0.2 %. However, even more recently, on 4 November 2019, the Italian Ministry of Health approved and ratified the “Definition of maximum levels of tetrahydrocannabinol (THC) in food” (GU n.11, 15-1-2020) [28]. This document fixes the content of THC at a maximum of 2 mg per kilo (0.0002%) in hemp seeds, flour, and derived foods and at a maximum of 5 mg per kilo (0.0005%) for the oil obtained from hemp seeds. It should be noted that the list of regulated foods provided in the appendix includes only seeds, flours and oil, but it seems that it will soon be updated based on new scientific evidence. Currently, the “Union method for the quantitative determination of the Δ9-tetrahydrocannabinol content in hemp varieties”, described by the annex III of the Commission Delegated Regulation (EU) 2017/1155 (last updated on 15 February 2017), is the only official procedure that member states must use for the quantitative determination of THC by gas chromatography (GC) after extraction with a suitable solvent [29]. It describes in detail everything about sampling, including sample dimensions, drying and storage, and techniques and reagents for the extraction and determination of THC, and it provides an allowed tolerance equal to 0.03% in absolute value. However, this official method is quite laborious, and the scientific community is always looking for advanced methodologies that will allow us to rapidly analyze natural mixtures without requiring manipulations or separations. An effective alternative to classical analytical methodologies could be the use of nuclear magnetic resonance (NMR) spectroscopy. NMR spectroscopy is a powerful and versatile technique that has progressively become a well-established tool in different areas of scientific research such as medicine, biology, and chemistry. The importance of NMR in the structural investigation of chemical compounds in liquid or solid phases is widely known, and its applicative power in addition to mass spectrometry (MS) has brought satisfactory results, particularly in the metabolic characterization of complex mixtures such as foods or natural extracts [30]. Despite it yielding relatively low-sensitivity measurements compared to MS (10 to 100 times better) with lower limits of detection (LOD), typically with an order of magnitude around micromolar [31], high-resolution NMR is becoming increasingly popular for fingerprinting as well as profiling. In particular, compared to MS, NMR spectroscopy is non-destructive, non-biased, non-invasive, does not damage analytes, and allows the use of samples such as tissues obtained, e.g., from biopsies, for further experiments [30]. Moreover, this technique is often fast and with low operating costs, it is easily quantifiable, and requires little or no chromatographic separation, sample treatment, or chemical derivatization. NMR is also a multinuclear technique that permits the routine and contemporaneous identification of a wide range of metabolites (such as sugars, organic acids, alcohols, polyols, and other highly polar compounds) in a highly quantitative and reproducible way thanks to 1D and 2D experiments. In addition, the combination of this high-throughput technique with chemometric methods is extremely advantageous because it gives the possibility to visualize, maximize, and therefore analyze the useful information contained in the experimental NMR data. The research presented in this work is placed in this scenario and aims to apply NMR methodologies to the study of natural extracts from the seeds and inflorescences of different cultivars of *C. sativa* with a THC/CBD << 1 ratio [32,33,34,35].

Specifically, the work aims to: (a) characterize the chemical profiles of the inflorescences and seeds for different varieties of *C. sativa* grown in Calabria (South of Italy) via NMR spectroscopy and, in particular, by using 1D (^1^H NMR, ^13^C NMR) and 2D (^1^H COSY, ^1^H-^13^C HMQC, ^1^H *J*-Res) experiments; (b) evaluate the extraction efficiency of different common solvents such as hexane, acetone and ethanol; (c) perform a multivariate statistical analysis (principal component analysis—PCA) based on 1D NMR data to discriminate samples coming from different extractive processes and/or varieties by identifying correlations between the metabolites that influence each metabolic profile; (d) and perform a quantification via NMR of the main cannabinoids (CBD, CBDA, and eventually Δ^9^-THC) using different internal standards.

## 2. Materials and Methods

### 2.1. Plant Material and Extraction Procedure

In this work, two different varieties of hemp were considered—*Tiborszallasi* and *Kompolti*—for both the metabolic characterization of inflorescences and for the quantitative and statistical analyses. NMR assignments were made on both varieties. Hemp inflorescences from *Tiborszallasi* and *Kompolti* grown in Calabria were harvested in September 2020, i.e., in the ripening period for both cases, selecting a reasonable number of plants for each cultivar located a few meters away in the same crop. All the collected hemp inflorescences were naturally air-dried, manually separated from twigs, and finely chopped. After this procedure, the samples were stored in the dark at 4.0 °C until analysis. Moreover, the chemical composition of the seed extracts was also investigated via NMR. The seed samples were collected from the *Futura 75* cultivar by selecting, as for the inflorescences, an appropriate number of plants representative of the entire crop. The dry seeds were ground into a powder and stored in the dark at 4.0 °C until analysis.

All inflorescence and seed samples were provided by “Calabria Maceri e Servizi S.p.A.” (Rende, CS, Italy), while the crops were produced by the farm “Le Querce S.r.l” (Montalto Uffugo, CS, Italy).

The storage, pre-treatment sampling, and extraction procedures for the inflorescences of the *Tiborszallasi* and *Kompolti* varieties were mostly in accordance with the official “Union method for the quantitative determination of the Δ^9^-tetrahydrocannabinol content in hemp varieties” (Annex III of the Commission Delegated Regulation (EU) No. 639/2014, 11 March 2014) [29]. In order to evaluate the efficiency of the extraction and thus highlight any differences between the various extracts, different extraction solvents commonly available in chemical laboratories and with increasing polarity were chosen. The solvents used were n-hexane, acetone, and ethanol. For each sample, 1.0 g of dried, chopped, and stored inflorescence was extracted with 45 mL of solvent at room temperature for 20 min using an ultrasonic bath (30 kHz frequency). The obtained extracts were centrifuged for 5 min at 3000 rpm, the solutions were paper filtered, and the residues were extracted once more using the same procedure with another 45 mL of the same solvent. Lastly, the solvents were completely removed under vacuum. Starting from the same dried inflorescence matrix, 24 extractions were carried out for each variety. For the *Tiborszallasi* variety, 9 extractions were performed using ethanol and acetone and 6 using hexane, while for the *Kompolti* variety, 9 extractions were performed using hexane and ethanol and 6 using acetone. For the quantitative analysis, for both varieties, each extraction was carried out in triplicate to calculate an average value for the extraction yield and estimate the relative error. Then, three samples were collected for each solvent for a total of nine extracts for each variety. In addition, a quantitative analysis with gas chromatography (GC) using the flame ionization detector (FID) method was conducted on samples of the *Tiborszallasi* variety prepared from the same dried inflorescence matrix as the NMR samples by following the protocol reported in the literature [29]. It should be emphasized that, given the chemical complexity of *C. sativa*, the extraction and collection of its various bioactive compounds are not simple and, for this reason, both solvents and different extraction methods are reported in the literature, ranging from microwave-assisted extraction to supercritical fluid extraction [36].

The extraction procedure for seeds of *Futura 75* were based on dynamic-maceration ultrasound-assisted extraction (UAE; Hielcher UP 100Hz, 100 W pulse, 30 kHz frequency), using ethanol as the solvent. Then, 2.00 g of seeds—dried, chopped and stored—were extracted with 45 mL of ethanol at room temperature for 20 min under magnetic stirring. The solution was then paper filtered, evaporated under vacuum at 30 °C, and the residue was extracted with the same procedure one more time with another 45 mL of same solvent [37]. The schematic experimental steps are shown in Figure 2.

### 2.2. Chemicals and Solvents

Pure solvents, ethanol (Absolute, ≥99.8%—VWR Chemicals, Briare, France), n-hexane (Laboratory Reagent, ≥95%—Sigma Aldrich, Darmstadt, Germany) and acetone, (ACS Reagent, ≥99.5%—Sigma Aldrich, Darmstadt, Germany) were used for the cannabinoid extractions. Deuterated chloroform (CDCl_3_—99.95 atom % D) as the solvent for NMR sample preparation, and anthracene, benzoic acid and 3-(trimethylsilyl) propionic-2,2,3,3-d_4_ acid (TMSP-d_4_—98 atom % D) as internal analytical standards for the quantitative NMR analysis were purchased from Sigma-Aldrich (Milan, Italy).

### 2.3. NMR Sample Preparation, Experimentation, and Data Processing 

To prepare the NMR sample, after the evaporation under vacuum, 30.0 mg of seed extract of *Futura 75* was dissolved in CDCl_3_ (~1 mL) directly in a 5 mm *o.d.* NMR tube. In this solution, the ^1^H NMR spectrum (spectral width (SW) of 14.00 ppm, 128 free induction decays (FIDs) and a relaxation delay of 5.0 s) and the 2D ^1^H COSY experiment (SW of 14.00 ppm on both dimensions, 2K data points, 40 scans, and 256 increments) were recorded. Two other similar extraction procedures were repeated on the same starting matrix of the dried seeds. From these extracts, 1D ^1^H NMR spectra were recorded to be used for reproducibility and standard deviation in the calculation of the essential fatty acids ratio. ^1^H NMR spectra were manually phased, baseline-corrected, and the chemical shifts were reported with respect to the TMS signal used as reference. From the ^1^H NMR spectra of these extracts, the main fatty acids ω-6/ω-3 ratio can be determined by combining the integrals, obtained after applying the deconvolution procedure, of three different signals: (a) the methyl protons of all the acyl groups (LA), with the exception of those of α-linolenic acid; (b) the methyl protons of ω-3 fatty acid (α-linolenic acid (αLA)); (c) the methylene protons of the linoleic and α-linolenic acyl groups; and using the relations [38]:(1)$$\alpha \mathrm{LA}=\frac{\left(b\right)}{\left(b\right)+\left(a\right)}$$
(2)$$\mathrm{LA}=\frac{3\cdot \left(c\right)-4\cdot \left(b\right)}{3\cdot \left[\left(b\right)+\left(a\right)\right]}$$

The extract residues of the two inflorescence varieties were dissolved in 1.20 mL of CDCl_3_ and 600 µL of this solution was transferred into a 5 mm *o.d.* NMR tube. For the quantitative analysis, samples of hemp in CDCl_3_ were prepared by carefully weighing all the components and by adding 0.3 mg of internal standard (anthracene, benzoic acid and TMSP-d_4_). No additional treatment was necessary for the preparation of the NMR samples [39,40].

All the NMR experiments were performed on a Bruker Avance 500 MHz spectrometer (Bruker, Fällanden, Switzerland) working at a field strength of 11.74 T (500 MHz ^1^H Larmor frequency), equipped with a 5 mm multinuclear probe TBO (triple-resonance broadband observe) and a standard variable-temperature control unit BVT-3000 (Bruker, Fällanden, Switzerland). All isotropic spectra were recorded at room temperature with CDCl_3_ used as a field-frequency lock signal.

Spectral assignments of metabolites were based on the one-dimensional (1D) ^1^H, ^13^C, and ^13^C-{^1^H} NMR spectra, the bi-dimensional (2D) homo- and heteronuclear correlation NMR experiments (^1^H COSY, ^1^H-^13^C HMQC) and by comparison with the data reports in the literature [39,40,41]. In addition, *J_ij_* couplings between some pairs of protons were measured thanks to the homonuclear 2D experiment of ^1^H *J*-Resolved spectroscopy (*J*-Res) [42,43]. For each ^1^H NMR experiment, 128 FIDs were acquired using a spectral width of 14.00 ppm and a relaxation delay of 5.0 s. The 1D ^13^C-{^1^H} NMR spectra were recorded with proton broad-band decoupling, collecting 8K FIDs using a SW of 250.00 ppm and a relaxation delay of 5.0 s. For an accurate quantitative analysis of the metabolites present in the complex mixture, it was necessary to calibrate both the 90° pulses on the monitored nuclei (^1^H and ^13^C) and the T1 spin-lattice relaxation time. The T1 relaxation time was estimated by using the conventional inversion recovery experiment (10 increments from 0.5 ms to 30.0 s for ^1^H, and 24 increments from 0.1 ms to 300.0 s for ^13^C) [44]. The ^1^H quantitative NMR (qNMR) spectra were recorded using the same acquisition parameters described before but with a relaxation delay of 20.0 s. Instead, for ^13^C qNMR, quantification experiments (*zgig* Bruker pulse sequence) were performed, collecting 4000 FIDs using a SW of 250.00 ppm, a relaxation delay of 160.0 s, and an acquisition time of 10.0 s. 

The initial relative quantification was obtained using Equation (3), in which the molar ratio $\frac{M_{X}}{M_{Y}}$ between the metabolites to quantify (X) and the internal standard (Y) is reported as a function of the ratio between their integral (I_X_ and I_Y_) and the ratio of resonant nuclei that generates the considered signal (N_X_ and N_Y_) [45,46].
(3)$$\frac{M_{X}}{M_{Y}}=\frac{I_{X}}{I_{Y}}\cdot \frac{N_{Y}}{N_{X}}$$

Then, on the basis of the mass of extract used to prepare the NMR sample and the relative extraction yield, the absolute quantification was obtained in terms of the percentage of the dry weight of the hemp flowers.

The ^1^H COSY experiments were acquired using a SW of 14.00 ppm in both dimensions, 2K data points, 40 scans, and 256 increments; the ^1^H-^13^C HMQC spectra were recorded using SWs of 14.00 ppm (^1^H) and 250.00 ppm (^13^C), 2K datapoints, 512 scans, and 40 experiments, and ^1^H *J*-Res spectra were acquired with a SW of 12.00 ppm, 2K datapoints, 256 scans, and 48 experiments. A *sine* and a *qsine* filter were applied in both dimensions, F1 and F2, for the COSY and HMQC experiments, respectively, before being Fourier-transformed. Then, 1D NMR spectra were Fourier-transformed and manually phased, baseline-corrected, and aligned using the TMS signal as a reference. ^13^C-{^1^H} NMR spectra were filtered with 1.0 Hz line broadening before Fourier transformation. For the multivariate statistical analysis, the ^1^H NMR spectra were segmented in a rectangular bucket fixed at 0.05 ppm. The integration region was defined in the first proton spectra and next it was reported, once saved, in the other spectra bucketed automatically. In particular, variables were manually selected by choosing the regions of NMR spectra with characteristic signals of metabolites and eliminating regions with a poor signal-to-noise ratio. Therefore, regions selected for the subsequent statistical treatment were 0.50–1.15 ppm, 1.3–2.6 ppm, and 3.8–7.0 ppm. Once normalized, the integrals were organized in a data matrix that was mean-centered and scaled. All the data processing steps were carried out using TopSpin 3.6 software (Bruker BioSpin, Rheinstetten, Germany) (TopSpin, 2018) [47].

### 2.4. Statistical Analysis: Principal Component Analysis (PCA)

Multivariate analysis was performed on the 48 1D ^1^H NMR experiments recorded from the samples prepared from the extracts obtained with different extractive solvents. There were 24 samples for both varieties: *Tiborszallasi* and *Kompolti*. It should be noted that each sample prepared came from a different extract, i.e., for example, the 9 samples of *Kompolti* in hexane were obtained from 9 extractive procedures of the corresponding inflorescences. Among the possible statistical approaches, in this work, principal component analysis (PCA) was used as an explorative method. PCA is a technique able to reduce the dimensionality of a multivariate problem without losing information. This mathematical treatment allows us to clearly visualize samples in a two- or three-dimensional space and reveals trends in the data or groupings of samples (clusters) based on their similarity [48,49,50]. This methodology, applied to the data matrix of the bucketing ^1^H NMR spectra of all extract samples, obtained as described in the previous subsection, allowed us to obtain two datasets of 24 samples and 130 variables for each variety that were used as starting points to carry out PCA analysis using R software (R software (R Core Team, 2019)) (Vienna, Austria) [51].

### 2.5. Chromatographic Experiments

GC-FID analysis was intended to verify the qNMR results for the *Tiborszallasi* variety. It was carried out by using the protocol reported in [29] and a gas chromatograph (GC) equipped with a split/splitless injector and a flame ionization detector (FID) (Dani Master GC1000, Dani instrument, Milan, Italy).

## 3. Results and Discussion

### 3.1. NMR Characterization of Seeds Extracts

Figure 3 shows the 1D ^1^H NMR spectrum of a sample of hemp seed extracts prepared as described in Section 2.1.

From the proton spectrum, it is possible to recognize the classic profile of the esters of polyunsaturated fatty acids (PUFAs) [52,53] and, from the 2D ^1^H COSY spectrum, it is possible to trace all the correlations of the triacylglycerols (TAGs) containing both saturated and unsaturated fatty acids, as shown in Figure 4.

Let us consider, for example, the strong signal generated by the α-methylenic protons of all acyl chains (H_A_, Figure 4) detected between 2.27 ppm and 2.37 ppm. Clearly, the resonance frequency of the group of nuclei (labelled in Figure 4) for all fatty acids was not exactly the same, and it depended on the nature of the chain to which they belonged. Consequently, it was not possible to determine a well-resolved multiplicity to the final complex signals appearing in the spectrum due to the overlap of all these signals with slightly different chemical shifts. For this reason, it was indicated as a multiplet (*m*) and the corresponding range of chemical shift is reported in Table 1, in which the correlations obtained from the COSY spectrum for these protons are also reported. The same reasoning is valid for all the other signals reported as multiplets in Table 1, which summarizes all assignments for hemp seed extracts obtained from the ^1^H and ^1^H COSY spectra.

Since the integrals of the ^1^H NMR signals were proportional to the number of hydrogen atoms present in each functional group and, overall, to the number of functional groups present in the sample, from the combination of the integrals of different signals it is possible to calculate the concentration of fatty acids in general, and the ω-6/ω-3 ratio in particular. To this end, three different signals in the protonic spectra were considered: (a) the multiplet at 0.88 ppm due to the overlapping triplet signals of the methyl protons of all the acyl groups (LA), with the exception of those of α-linolenic acid; (b) the triplet at 0.97 ppm generated by the methyl protons of ω-3 fatty acid (α-linolenic acid; (αLA)); (c) the multiplet at 2.72–2.86 ppm generated by the diallylic protons of the linoleic and α-linolenic acyl groups. By combining the area of these signals, using the relations (1) and (2) that take into account the number of equivalent nuclei in each group, the concentrations of αLA and LA were calculated, from which the ω-6/ω-3 ratio was obtained [38].

Calculations were made considering three different samples for reproducibility and to give a main value and a standard deviation. The value obtained for the ω-6/ω-3 ratio using Equations (1) and (2) was 2.93 ± 0.07. As can be seen, this value obtained by ^1^H NMR is very close to the value of 3:1 for ω-6/ω-3 considered optimal for human dietary purposes, since it is able to prevent various diseases such as diabetes, cardiovascular disease, cancer, and other chronic diseases [4,9,53,54]. On the other hand, the growing interest in hemp seed oil in other fields such as pharmaceuticals and cosmetics [53] has resulted in a constant search for methods that allow fast and systematic quality control: the ω-6/ω-3 ratio is one of these quality parameters [55]. Our result is doubly important because, on the one hand, it indicates the quality of the oil extracts from the seeds of the *Futura 75* cultivar grown in Calabria and, on the other hand, it confirms that NMR is a reliable quantitative platform for the fast screening of hemp oil quality. Indeed, the measurement of this ratio is based only on the recording and analysis of the ^1^H NMR spectra obtained directly from the seed extracts without further derivatization, as is required by the gas chromatography (GC) method, the common and validated method used to determine the composition of oil in terms of fatty acids [56]. Moreover, our result agrees with that reported in the paper of Siudem et al. [38] in which the authors analyzed six different samples of hemp seed oils and calculated the ω-6/ω-3 ratio by ^1^H NMR for each of them using the relationships (1) and (2). The authors compared these data with those from CG method applied to the same hemp seed oils and found a substantial agreement between them, which proves the effectiveness of the method. It is worth noting that, in a recent paper [53], the ^1^H NMR methodology—as well as being used in the evaluation of the ω-6/ω-3 ratio—is successfully combined with chemometric methods in order to observe the differences in several oil samples due to the different time and storage conditions of the oils. Hence, once again, this demonstrates how NMR can be used for the rapid and reliable analysis of hemp seed oil quality as an alternative to the more common classical analytical methods.

### 3.2. NMR Characterization of Flower Extracts

Figure 5 shows the ^1^H and ^13^C NMR spectra of inflorescence ethanolic extracts for the *Tiborszallasi* variety. As is evident from the figure, the two spectra exhibit a complex distribution of resonances due to strongly overlapped signals of cannabinoids with similar molecular structures (Figure 1), from which it is very difficult to recognize single metabolites through simple inspection.

To overcome these limitations that are characteristic of the 1D spectra of complex mixtures, the identification of the various cannabinoids present in the extracts (CBD, CBDA, CBG, THC) was carried out using the 2D correlation experiments ^1^H COSY, ^1^H-^13^C HMQC, and ^1^H *J*-Res and the data reported in the literature regarding the NMR characterizations of many single isolated cannabinoids [40,41,42]. It is worth underlining that, even in recent papers [35,36,57], the NMR methodology is mostly used to characterize fractions or single components of *C. sativa* obtained by separation techniques. In Figure 6, ^1^H COSY and ^1^H-^13^C HMQC spectra are shown together with the corresponding enlargement on the CBD and CBDA correlations taken as an example of the metabolic identification procedure adopted. In order to fully characterize the inflorescence extract, the 2D ^1^H *J*-Res experiments were performed on the same sample [42,43]. This type of experiment is capable of separating coupling and chemical shift information into two orthogonal dimensions, allowing for multiplet analysis. The one-dimensional spectrum (^1^H or ^13^C) is shown in the F2 dimension, while the couplings related to each chemical shift can be read in the indirect dimension, F1, at the chemical shift value of the multiplet signal to be studied. In Figure 7, the 2D ^1^H *J*-Res spectrum recorded on the sample of *Tiborszallasi* is reported, and, as an example, the case of H-9_cis_ signal of CBD is reported in the enlargement. The signal of this proton in the 1D ^1^H NMR spectrum is a multiplet from which it is impossible to extract any useful information. On the contrary, the extrapolation of the column in correspondence to the chemical shift of H-9_cis_ in the F2 dimension of the ^1^H *J*-Res spectrum gives the 1D profile of the considered signal in which it is possible to measure all the coupling constants. Indeed, as can be seen from the enlargement of Figure 7, this signal is a doublet of a quadruplet due to the interaction of the proton H-9_cis_ with the proton H-9_trans_ (doublet: ^2^*J_9cis-trans_* = 2.6 Hz) and with the methyl proton H-10 (quadruplet: *^4^J_9cis-10_* = 0.9 Hz).

The assignments for the various cannabinoids present in the ethanolic extract, CBD, CBDA, and CBG of the hemp inflorescences of the *Tiborszallasi* variety and their relative experimental information obtained from the NMR spectra (1D and 2D) are summarized in Table 2. These results agree with many other literature data [35,39,40,41] and demonstrate how the non-separative NMR technique, combining 1D and 2D experiments, can be successfully applied directly to a complex mixture, such as a *Cannabis* extract, for chemical characterization. An interesting result of the present study is the measurement of a large number of J_H-H_ for the main cannabinoids, carried out again directly on the *Cannabis* extract through the analysis of the 2D J-Res spectra. It must be highlighted that, in the ethanolic extract, the signals related to THC were not detected in the recorded spectra; this means that, in this solvent, the quantity of THC extracted was below the NMR detection limit [31]. This also occurred for the acetone extracts while the spectra of the hexane extracts showed a small broad peak isolated at 6.40 ppm corresponding to the proton H-10 of Δ^9^-THC. This means that hexane, which is also the solvent indicated in the official method for determining the amount of Δ^9^-THC in hemp, was the most effective at extracting cannabinoids compared to the other solvents used. Once the assignment of cannabinoids in the *Tiborszallasi* variety was determined, a comparison between the NMR spectra of the samples relating to the *Kompolti* variety was possible. Except for the proton spectrum of the hexane extract of the *Kompolti* variety, in which Δ^9^-THC signals were absent, no further differences in terms of profile were distinguished in the spectra of any other samples of either variety. All these spectra are reported in the Appendix A, where it is also possible to find the comparison between ^1^H NMR spectra from both varieties for the different solvents.

In addition to the cannabinoids, by comparison of the protonic spectra of the hemp seed oil and hemp inflorescence extracts, it was possible to recognize in this last spectrum some signals that referred to the triacylclycerols constituent. In particular, this profile was easily recognizable in the samples obtained by using ethanol as the extracting solvent. All the signal assignments are reported in Figure 8. Regarding the solvents, it must be underlined that many are used for cannabinoid extraction but, to date, there is no specific protocol. However, it is not surprising that we were able to better recognize the fatty acid profiles in the ethanol extracts since, from the literature data [58], it seems that ethanol has a greater extraction power than acetone and hexane. On the other hand, hexane, while showing the worst performance in terms of total yield, leads to cleaner extracts with fewer contaminants and extracts richer in cannabinoids [59]. For this reason, hexane is used for the selective analysis of cannabinoids in the official method established by the European Commission [29]. It should be noted that this was also evident in our spectra recorded on the hexane extracts. Indeed, these extracts have a cleaner profile than the other two solvents and allowed us, for the Tiborszallasi variety, to quantify the THC content together with the other cannabinoids.

### 3.3. Multivariate Analysis

Principal component analysis (PCA) was applied to discriminate hemp flowers samples based on their ^1^H NMR spectra [48,49,50]. The main aim was to observe how the cannabinoid profiles changed for extracts in relation to the nature of solvent extraction and the efficiency of the extraction procedure. PCA was carried out on two different data matrices, one for each hemp variety. For *Tiborszallasi,* the data matrix consisted of 24 samples (9 extracts for ethanol and acetone, respectively, 6 extracts for hexane) and 130 variables. For the *Kompolti* variety, the data matrix consisted of 24 sample (9 extracts for hexane and ethanol, respectively, 6 extracts for acetone) and 130 variables.

Figure 9 and Figure 10 show the 3D score plot for the *Tiborszallasi* and *Kompolti* variety samples, respectively, with a cumulative percentage of explained data variance for the three first PCs equal to: (a) 88.1 % (45.5% for PC1, 30.2% for PC2, 12.4% for PC3) for the *Tiborszallasi* variety and (b) 84.5% (51.8% for PC1, 17.5% for PC2, 15.2% for PC3) for the *Kompolti* variety. This means that data loss was negligible in both cases. The score plot effectively summarizes the relationship between the samples and highlights what was not discovered by the simple comparison among the protonic spectra: clear and well-defined separation between samples was observed and every type of extract was clustered into one defined region. Indeed, PC1 allows us to discriminate among extracts with non-polar solvent and extracts with higher polarity: hexane extracts had, in fact, positive values of PC1, while the other ones had negative values of PC1. Instead, PC2 seems to best discriminate between the acetone and ethanol extracts. The same result was obtained for both varieties.

However, in order to determine which metabolites have greater influence on the discrimination and evaluation of the quality and efficiency of the extraction procedure, an accurate discussion about loadings is necessary.

In Figure 11, a 2D biplot of the first two PCs (PC1 and PC2) for the *Tiborszallasi* variety is reported; this shows the sample differentiation and the changes in the metabolite concentrations from one extract to another and, consequently, the variables responsible for the sample clustering can be observed in the score diagram.

THC is the marker that gives the first clear distinction between hexane and all the other extracts: the THC loadings had positive values of PC1 and negative values of PC2, and these were very close to the scores of the hexane extracts. This means, as also demonstrated by the loadings of CBD and CBDA, that hexane is more efficient in the extraction of cannabinoids and of THC in particular. Indeed, the latter was below the NMR detection limit in the acetone and ethanol extracts. These results are not surprising since it is well known from the literature [34,60] that the polarity of the solvent affects the chemical composition of the cannabinoids present in the extracts.

Focusing on the TAGs previously detected during the signal assignment step, negative values of PC1 were obtained for these loadings. This indicates that extraction with a higher polarity solvent (acetone or ethanol) obtains samples with a lower percentage of cannabinoids but that are richer in fatty acids than hexane extracts. This peculiarity could be exploited to obtain extracts rich in both bioactive compounds—cannabinoids and TAGs—that could potentially be used in the food and pharmaceutical industries to produce functional foods and supplements.

The same conclusions were also reached for the *Kompolti* variety, whose 2D biplot of the first two PCs is reported in the Appendix A.

The present study has shown that the exploratory PCA method, although simple, is able to differentiate samples coming from different extraction solvents and, in addition, it highlights which cannabinoids form the basis of this differentiation. Chemometrics-aided NMR techniques performed on *C. sativa* components (inflorescences, seed oils, leaves, and other less valuable parts of the plant), or on fractions of them, are widely used to determine different properties of *C. sativa,* as evidenced by the plethora of studies reported in the literature [34,41,61,62,63,64,65] and the references reported therein. In many of these works, the combination of ^1^H-NMR spectra with chemometric tools was used for the metabolomic differentiation of inflorescence extracts from different cultivars and to identify particular markers in order to discriminate different plant chemotypes [41,61]. In other studies [62,66], targeted and non-targeted NMR methodologies were used to identify and quantify compounds of different classes present in the inflorescence extracts of several *C. sativa* cultivars and to monitor their variations in three different harvested stages. The cultivars analyzed in these studies had a THC content always below the legal limit, while the quantities of the other cannabinoids in the extracts were affected by the harvest time and by the solvent. Although it is not easy to make comparisons given the quantity of factors affecting the composition of cannabinoids, the results obtained in this study substantially agree with the studies described above, and therefore seem to demonstrate the reliability of this method and its possible application in the routine analysis of cannabinoids.

### 3.4. Quantitative Analysis of Inflorescences

As a support to the results obtained from the PCA analysis and to evaluate if the NMR methodology could be a valid tool in the quantitative determination of the cannabinoids extracted as a function of the solvent used, an NMR quantification was also carried out using ^1^H and ^13^C NMR spectra and different internal standards.

*Tiborszallasi-* and *Kompolti*-type hemp flower samples used for the quantification were obtained, as mentioned before, by ultrasound-assisted extraction using three different common solvents of increasing polarity: hexane, acetone and ethanol. The complete procedure, from extraction to NMR measurement, was performed in triplicate to evaluate repeatability, to calculate an average value for the extraction yield, and to estimate the relative error. The procedure below is described for the *Tiborszallasi* samples, but exactly the same was performed for the *Kompolti* variety.

The average experimental extraction yields for each procedure were: (a) 17.9 ± 0.3% for extraction with hexane; (b) 9.9 ± 0.8% for extraction with acetone; (c) 19.6 ± 0.9% for extraction with ethanol.

For the quantification of the cannabinoids present in the extracts, both 1D ^1^H NMR and ^13^C NMR spectra were used. The quantification of cannabinoids by protonic spectra included the use of three different internal standards: anthracene, in accordance with literature [20], benzoic acid, and 3-(trimethylsilyl)propionic-2,2,3,3-d_4_ acid (TMSP-d_4_). These compounds are highly pure (≥ 99.9%), have low volatility, are chemically inert, and are not similar in structure to the cannabinoids to be quantified. Indeed, in the case of NMR spectroscopy, it is necessary that they generate well-isolated signals in the spectrum that do not overlap with the peaks assigned to other metabolites in the mixture. In this case, signals of aromatic protons of anthracene and benzoic acid ranged between 7.4 ppm and 8.5 ppm in the protonic spectrum, and the singlet of the methyl group of TMSP-d_4_ was set to 0.00 ppm.

The signals considered for the quantification, using Equation (3), were the following: for CBD, the H-9_trans_ olefinic proton signal was set at δ = 4.65 ppm; for CBDA, the H-9_cis_ olefinic proton signal was set at δ = 4.38 ppm; and for Δ^9^-THC, the H-10 proton signal was set at δ = 6.40 ppm. The results obtained for the three standards used are reported in Table 2. For further confirmation of the ^1^H qNMR results, quantitative ^13^C NMR spectra were also recorded on the hemp flower hexane extracts of *Tiborszallasi,* optimizing the acquisition parameters (see Section 2.3) and the operative conditions [40]. However, in this case, only TMSP-d_4_ was used as an internal standard for ^13^C qNMR because the aromatic carbons of anthracene and benzoic acid generated signals in the region between 125 ppm and 134 ppm, thereby overlapping with the cannabinoid signals. Moreover, given the low sensitivity of the ^13^C nucleus, the ^13^C qNMR was useful for the quantitative determination of CBDA, the only cannabinoid whose signals had an acceptable signal-to-noise ratio. The CBDA content (% on dry weight) obtained via ^13^C qNMR for *Tiborszallasi* in hexane extract was equal to 6.2 ± 0.9, which was substantially in agreement with that obtained from protonic spectra using the same internal standard (Table 2).

Observing the results reported in Table 2, the value obtained for the three cannabinoids CBD, CBD and Δ^9^-THC, for each type of extract using different internal standards, were fairly reproducible. This means that benzoic acid and TMSP-d_4_ seem to be valid alternatives to anthracene as internal standards for quantification. Indeed, anthracene is the standard that is usually used for the quantification of cannabinoids via NMR, but it has many drawbacks related to its toxicity, its poor solubility in common solvents, and long recording times due to long T1 [67]. Furthermore, the data show that hexane extracts are richer in cannabinoids than other solvents, as had already emerged in the biplot analysis, and, most interestingly, the quantities obtained for these extracts via NMR matched very well with the experimental data acquired following the official European procedure for the determination of cannabinoid Δ^9^-THC, as reported in Table 3. Indeed, the table shows the comparison, for the same sample of *Tiborszallasi* inflorescences, between the qNMR data of Δ^9^-THC via NMR and those obtained with the official GC-FID technique [29]. In the table, the GC-FID value concerning the quantity of CBD/CBDA obtained from the same sample is also reported. It should be noted that the GC-FID procedure does not allow discrimination between CBDA and CBD: before extraction, it includes a pretreatment of inflorescences at a high temperature that leads to the complete decarbossilation of the acidic form of this cannabinoid (CBDA) to the neutral (CBD). So, this unique value should be considered as the sum of the quantities of CBDA and CBD initially present in flowers. As can be seen from the table, also in this case, the data via NMR are in perfect agreement with those obtained via GC-FID. It is worth noting that the NMR technique has been proven to be a reliable and powerful tool for the quantification of different natural products and, especially in recent years, has also been successfully applied to the quantification of CBD and other cannabinoids directly on hemp extracts coming from different cultivars using both the ^1^H and ^13^C qNMR methodologies. [40,44,68,69,70,71]. Quantitative ^1^H-NMR is the most widely used method in the quantification of natural extracts and has been shown to have a good level of accuracy and reproducibility. However, its application to hemp extracts presents several drawbacks, due both to the presence of contaminants and to the overlapping of different signals that require the extensive use of deconvolution processes [72]. The use of ^13^C q-NMR, introduced by Marchetti et al. [40], partially removes these weaknesses and at the same time offers sufficiently precise and sensitive results. As expected, our results substantially agree with these previous reports on the ^1^H and ^13^C qNMR investigations, even if it is necessary to point out that a direct comparison on the quantities of cannabinoids found in the different extracts of the cultivar analyzed is practically impossible given, as we have already highlighted, the large quantity of variables, i.e., cultivar, geographical origin, harvesting period, agronomic practices, extraction methodologies, etc., that affect the composition of cannabinoids. However, the quantitative results of the present study highlight, once again, the remarkable potentialities of the NMR technique which was able to quantify the main metabolites present in the hemp inflorescence extracts we analyzed as they were, without the further treatment or derivatization required by the official technique [73,74]. Moreover, as reported recently by Dadiotis et al. [75] concerning the quantitative analysis of cannabinoids in hemp extracts using, in a complementary way, the ^1^H-NMR and ^1^H-^1^H COSY NMR spectra, these quantitative data via NMR are comparable with those acquired with other more consolidated techniques applied to the same extracts. This evidence shows good correspondence between the various quantification techniques, as also confirmed by the data we obtained on the *Timborszallase* cultivar given the satisfactory agreement between the NMR and GC-FID data of the hexane solvent.

In addition, the quantitative data obtained for the different solvents confirm the purely qualitative indications given by the PCA analysis, which proved to be very informative and fast.

These results were obtained for the *Tiborszallasi* variety but, as previously mentioned, the quantification of the main cannabinoids by ^1^H and ^13^C qNMR was also performed for the *Kompolti* variety and the results are reported in the Appendix A.

## 4. Conclusions

*Cannabis sativa* is a fast-growing plant currently grown all over the world that is gaining popularity in various fields of research for its biological and pharmaceutical properties. Actually, *C. sativa* is widely recognized and appreciated for the high nutritional and health-promoting properties of the oil obtained from its seeds, together with the pharmacological activity mainly associated with psychoactive and non-psychoactive cannabinoids and the chemical components mainly extracted from the inflorescences. In this work, NMR spectroscopy was applied to analyze extracts from the seeds and inflorescences of different varieties of *Cannabis sativa* grown in Calabria in order to explore the potentialities of this technique for the qualitative and quantitative analysis of the extracts, and to evaluate the possibility of using it as an alternative to the most common methods in the quantification of cannabinoids present in inflorescence extracts. The quantitative NMR results obtained from two varieties of hemp inflorescence extracts, using different internal standards and solvents, demonstrated the high potentiality of the proposed technique in this field of application. Indeed, the NMR technique was able to quantify the main cannabinoids present in the extracts, the quantitative data were reproducible, and—most importantly—the data from the hexane solvent were congruent with the data obtained by the GC-FID method. Moreover, while this last methodology is not able to distinguish CBD and CBDA, using the NMR method, it was possible to separate the two contributions and quantify them. This proves, once again, the analytical power of the NMR technique which is not only able to offer the same results obtained from the official method, including the evaluation of THC, but can indicate more informative data without performing particular treatments on the sample.

In addition to the characterization and the quantitative study, different extraction procedures were tested and evaluated by NMR spectroscopy with the aim of obtaining inflorescence extracts poor in psychotropic agents and rich in medical cannabinoids and triacylglicerols (TAGs), which have an ω-6/ω-3 ratio that has been found to be excellent from a nutritional point of view. Specifically, extracts of inflorescences obtained by ultrasound-assisted solute–solvent extraction using hexane, acetone and ethanol as solvents were studied. By elaborating the spectral data with a statistical method (PCA) together with the qNMR approach, it was possible to conclude that hexane was more efficient in the extraction of cannabinoids (THC included) than the TAG constituents, while extraction with a higher polarity solvent (acetone or ethanol) obtained samples free from THC (THC content < LOD), rich in TAGs, and with a lower percentage of cannabinoids. This evidence can be exploited to obtain extracts rich in bioactive compounds (both cannabinoids and TAGs) that could potentially be used in the food and pharmaceutical industries, opening new paths for the production of functional foods and supplements.

## Figures and Tables

**Figure 1 molecules-27-03509-f001:**
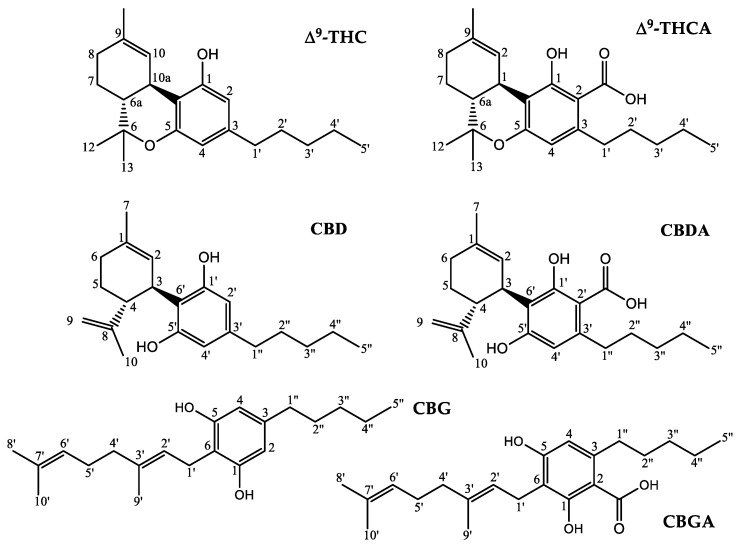
Chemical structure and nuclei numbering of molecular fragments in hemp principal cannabinoids.

**Figure 2 molecules-27-03509-f002:**
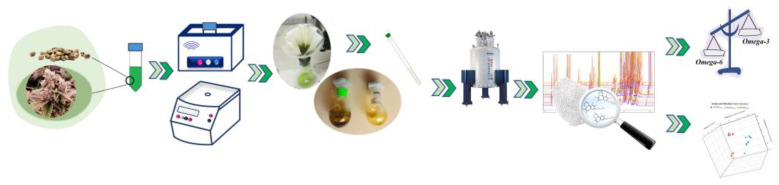
Schematic experimental steps involved in the extraction, sample preparation, and NMR characterization of *C. sativa* seeds and inflorescences.

**Figure 3 molecules-27-03509-f003:**
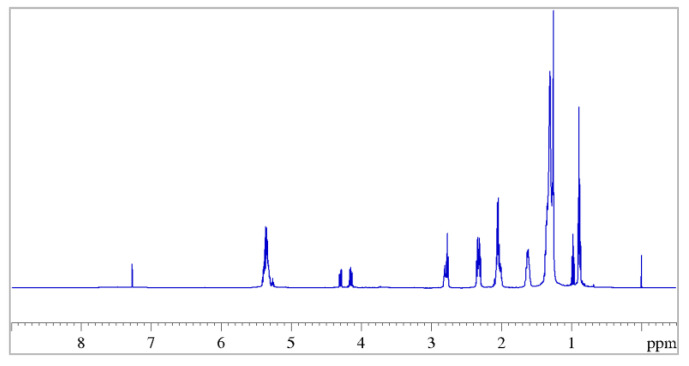
^1^H NMR spectrum (500 MHz) of hemp seed oil dissolved in CDCl_3_ recorder at 298K, obtained with ultrasound-assisted extraction (UAE) procedure.

**Figure 4 molecules-27-03509-f004:**
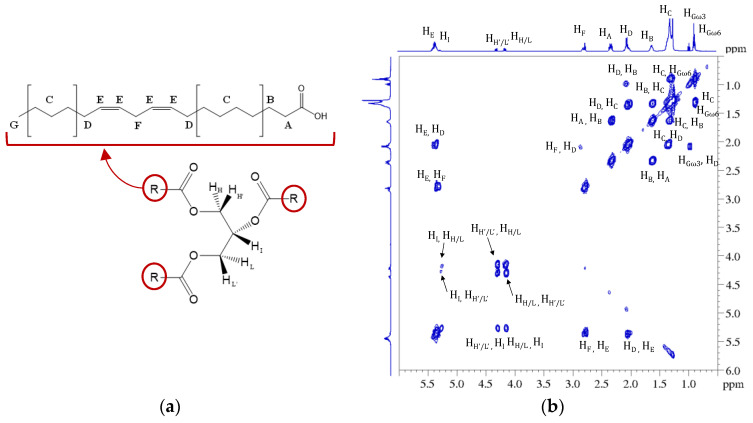
(**a**) General structure and nuclei labelling of molecular fragments in triacylglicerols (TAGs); (**b**) 2D map ^1^H COSY of the ethanolic extract with the UAE of *C. sativa* seeds. In the figure, all cross peaks corresponding to the homuncular correlations have been highlighted.

**Figure 5 molecules-27-03509-f005:**
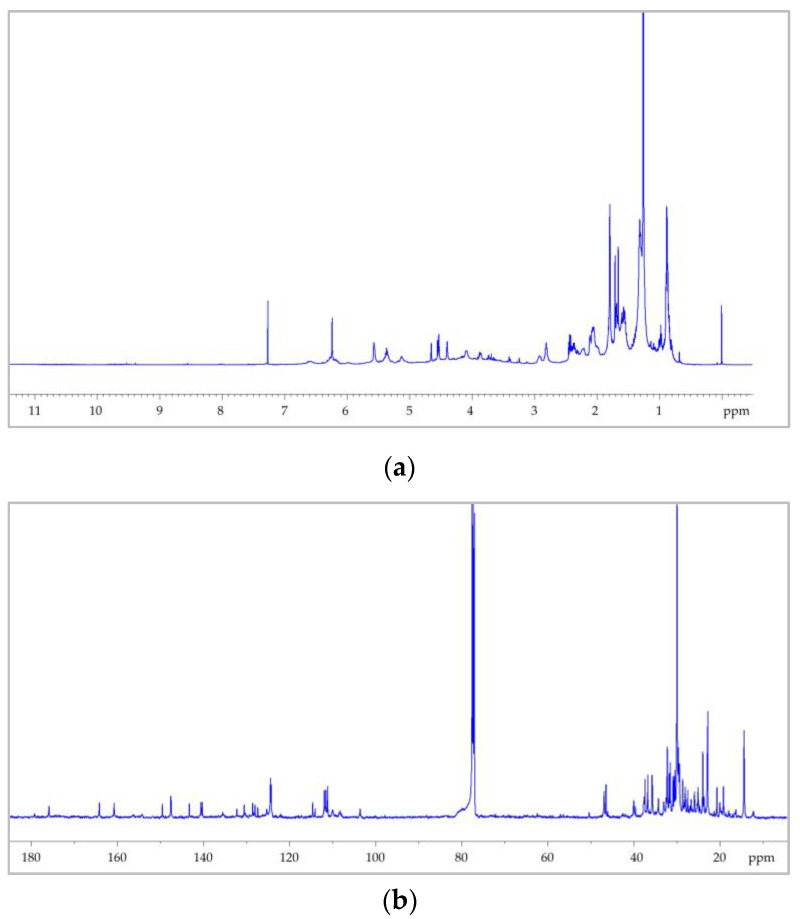
NMR spectra (500 MHz) of inflorescence extracts dissolved in CDCl3 recorded at 298K. (**a**) ^1^H NMR spectrum recorded using *zg30* Bruker standard pulses sequence; for each experiment, 128 FIDs were accumulated using a spectral width of 14.00 ppm and a relaxation delay of 5 s. (**b**) ^13^C-{^1^H} NMR spectrum (*zgig* Bruker pulse sequence) performed with proton broad-band decoupling, collecting 8K free induction decays (FIDs) and using a spectral width of 250.00 ppm and a relaxation delay of 5s. One-dimensional NMR FIDs were Fourier-transformed, phased, baseline-corrected, and aligned using the TMS signal as a reference. ^13^C-{^1^H} NMR spectra were filtered with 1 Hz line broadening before Fourier transformation.

**Figure 6 molecules-27-03509-f006:**
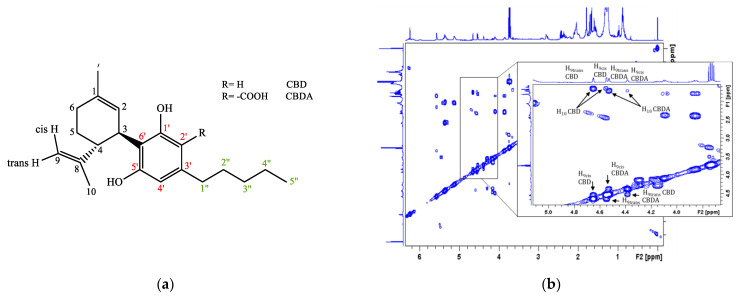
(**a**) Structure of CBD and CBDA reported together with atom numbering adopted; (**b**) 500 MHz ^1^H COSY spectrum; and (**c**) ^1^H-^13^C HMQC NMR spectrum of inflorescence ethanolic extract sample of *Tiborszallasi*, a variety dissolved in CDCl_3_ and recorded at 298K.

**Figure 7 molecules-27-03509-f007:**
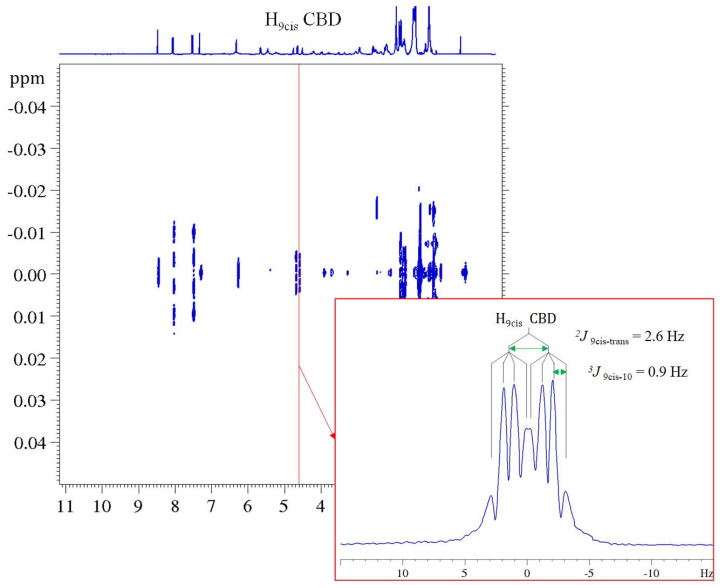
^1^H *J*-Res NMR spectrum of inflorescence ethanolic extract samples of *Tiborszallasi* dissolved in CDCl_3_ and recorded at 298K. The projection of the H-*9_cis_* signal (doublet of quadruplets) of CBD is reported in the enlargement.

**Figure 8 molecules-27-03509-f008:**
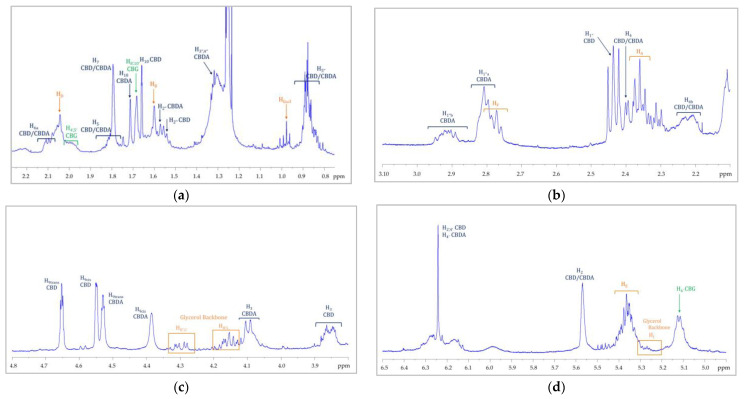
^1^H NMR spectrum of an ethanolic extract of *C. sativa* inflorescences (*Tiborszallasi* variety). For the enlarged regions of (**a**) [0.8 ppm–2.2 ppm], (**b**) [2.2 ppm–3.00 ppm], (**c**) [3.7 ppm–4.8 ppm], and (**d**) [4.8 ppm–6.5 ppm] the signal attributions are shown. Signal assignments are reported with different colors: orange for TAGs; grey for the glycerol backbone; blue for CBD/CBDA; green for CBG/CBGA.

**Figure 9 molecules-27-03509-f009:**
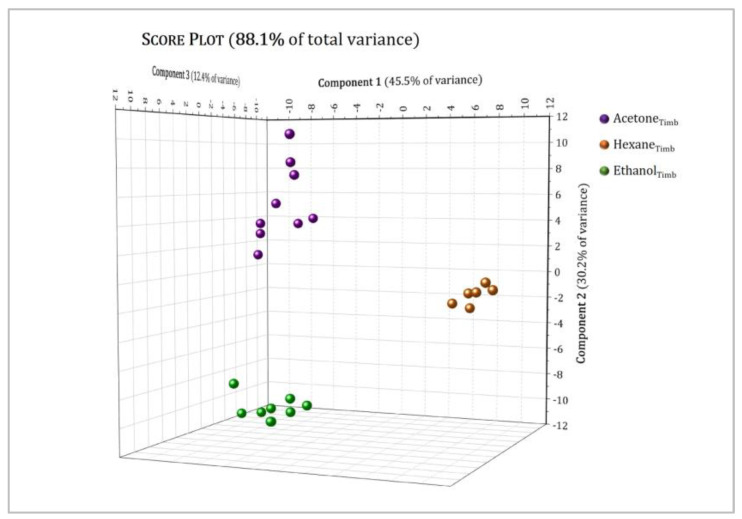
Principal component analysis (PCA) of hexane (brown dots), acetone (purple dots) and ethanol (green dots) extracts of the *Tiborszallasi* variety of hemp. The score plot shows the first three PCs (PC1, PC2 and PC3) with their respective variations. R2X(PC1) = 45.5%, R2X(PC2) = 30.2%, R2X(PC3) = 12.4%.

**Figure 10 molecules-27-03509-f010:**
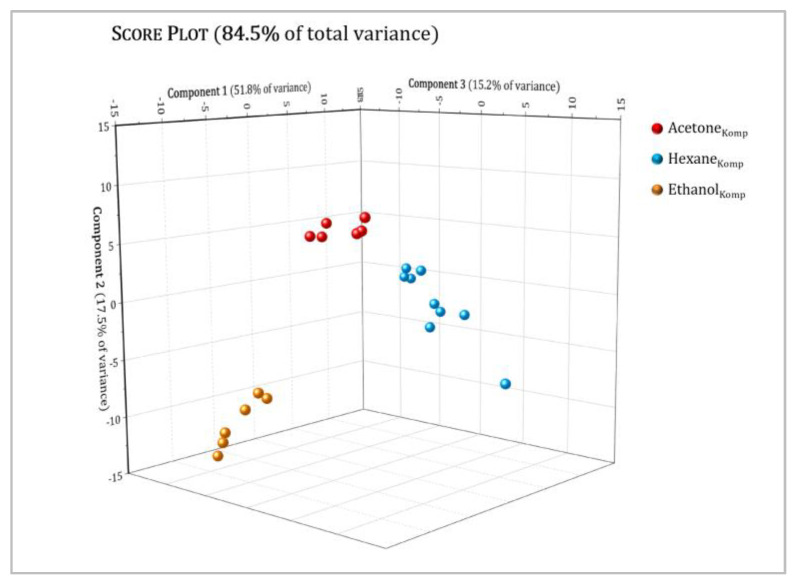
Principal component analysis (PCA) of hexane (blue dots), acetone (red dots), and ethanol (orange dots) extracts for the *Kompolti* variety of hemp. The score plot shows the first three PCs (PC1, PC2 and PC3) with their respective variation. R2X(PC1) = 51.8%, R2X(PC2) = 17.5%, R2X(PC3) = 15.2%.

**Figure 11 molecules-27-03509-f011:**
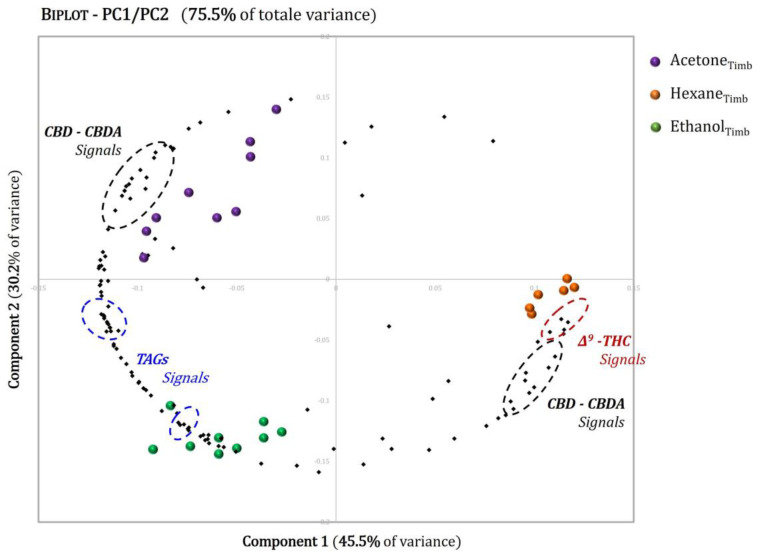
Biplot of PCA carried out on NMR spectra of hexane (brown dots), acetone (purple dots) and ethanol (green dots) extracts of the *Tiborszallasi* variety of hemp. The score plot shows the first two PCs (PC1 and PC2) with their respective variations. R2X(PC1) = 45.5%, R2X(PC2) = 30.2%.

**Table 1 molecules-27-03509-t001:** ^1^H NMR chemical shifts and ^1^H/^1^H correlations of fatty acid protons in triacylglicerols (TAGs) in CDCl_3_ for hemp seed extracts.

Position	δ_H,_ Multiplicity *^a^* (*J* in Hz)	COSY
A	2.27–2.37, m	E
B	1.55–1.67, m	C, A
C	1.23–1.39, m	G_ω6_ B, D
D	1.98–2.11, m	G_ω3_, C, F, E
E	5.28–5.42, m	D, F
F	2.72–2.86, m	D, E
G_ω3_	0.97, t	D
G_ω6_	0.88, m	C
H, L (*Gly ^a^*)	4.14, dd (11.88, 5.93)	H’, L’, I
I (*Gly ^a^)*	5.26, m	H, H’, L, L’
H’, L’ (*Gly ^a^)*	4.29, dd (11.88, 4.31)	H, L, I

*^a^* Abbreviations: *d*—doublet; *dd*—doublet of doublet; *t*—triplet; *m*—multiplet; *Gly*—Glycerol.

**Table 2 molecules-27-03509-t002:** ^1^H and ^13^C chemical shifts of the main cannabinoids in the flower extracts of *Cannabis sativa* (*Tiborszallasi* variety) in CDCl_3_.

Compound	δ ^1^H ppm (Multiplicity *, ^1^H-^1^H *J*-Coupling—Hz)		δ ^13^C ppm	
**CBD**	H_3_	3.86 (ddt; *J*_H3-H4_ = 13.00 Hz (d), *J*_H3-H2_ = 3.51 Hz (d), *J*_H3-H5_ = 2.51 Hz (t))	C_3_	37.01
	H_2_	5.57	C_2_	124.14
	H_6a_	2.05–2.09	C_6_	30.36
	H_6b_	2.22		
	H_5_	1.78–1.84 (ddd; J_H5-H4_ = 5.30 Hz (*d*), J_H5-H6a_ = 1.30 Hz (d), J_H5-H6b_ = 0.60 Hz (d))	C_5_	28.35
	H_4_	2.40 (dd; J_H4-H3_ = 13.00 Hz (d), J_H4-H5_ = 5.00 Hz (d))	C_4_	46.16
	H_7_	1.79 (d;^3^*J*_H7-H2_ = 0.50 Hz)	C_7_	23.69
	_H9trans_	4.64 (dq; *J*_9trans-9cis_ = 2.65 Hz (d), ^3^*J*_9trans-10_ = 1.50 Hz (q))	C_9_	110.81
	H_9cis_	4.53 (dq; *J*_9cis-9trans_ = 2.65 Hz (d), ^3^*J*_9cis-10_ = 0.92 Hz (q))		
	H_10_	1.66 (dd; ^3^*J*_10-9cis_ = 0.92 Hz (d), ^3^*J*_10-9trans_ = 1.50 Hz (d))	C_10_	20.30
	H_2′_	6.26	C_2′_	109.56
	H_4′_	6.16	C_4′_	107.92
	H_1″_	2.43 (t)	C_1″_	35.46
	H_2″_	1.52–1.61	C_2″_	30.65
	H_3″_, H_4″_	1.27–1.32	C_3″_	31.48
			C_4″_	22.54
	H_5″_	0.86–0.88	C_5″_	14.04
**CBDA**	H_3_	4.08	C_3_	35.38
	H_2_	5.55	C_2_	124.14
	H_6a_	2.05–2.09	C_6_	30.36
	H_6b_	2.22		
	H_5_	1.79 (ddd; J_H5-H4_ = 5.30 Hz (*d*), J_H5-H6a_ = 1.30 Hz (d), J_H5-H6b_ = 0.60 Hz (d))	C_5_	28.35
	H_4_	2.40 (dd; *J*_H4-H3_ = 13.00 Hz (d), *J*_H4-H5_ = 5.00 Hz (d))	C_4_	46.45
	H_7_	1.79 (d; *^3^J*_H7-H2_ = 0.50 Hz)	C_7_	23.69
	H_9trans_	4.51 (dq; ^3^*J*_9cis-9trans_ = 3.00 Hz (d); ^3^*J*_9trans-10_ = 1.76 Hz (q))	C_9_	111.21–111.25
	H_9cis_	4.39 (dm; ^3^*J*_9cis-9trans_ = 3.00 Hz (d))		
	H_10_	1.70	C_10_	18.91
	H_4′_	6.21	C_4′_	111.21–111.25
	H_1″a_	2.81	C_1″_	36.68
	H_1″b_	2.92		
	H_2″_	1.52–1.61	C_2″_	31.24
	H_3″_, H_4″_	1.27–1.32	C_3″_	31.94
			C_4″_	22.54
	H_5″_	0.86–0.88	C_5″_	14.04
**CBG**	H_2_	6.24	C_2_	108.25
	H_5′,_ H_4′_	2.04	C_4′_	32.28
			C_5″_	26.51
	H_6′_	5.12	C_6′_	125.08
	H_8′,_ H_10′_	1.68	C_8′_	20.51
			C_10″_	23.44

* Abbreviations: d—doublet; t—triplet; q—quadruplet; m—multiplet; dd—doublet of doublet; ddd—doublet of doublet of doublet; ddt—doublet of doublet of triplet; dq—doublet of quadruplet; dm—doublet of multiplet.

**Table 3 molecules-27-03509-t003:** ^1^H NMR data of the main cannabinoids in *Timborzallasi* inflorescences compared with the GC-FID method.

Compound	qNMR on Flowers UAE Extracts				GC-FID
	qNMR IS	Hexane	Acetone	Ethanol	
**CBDA** content *	Anthracene	6.3 ± 0.8	0.40 ± 0.1	0.31 ± 0.04	Validated Laboratory Method 6.9 ± 0.2 Referred to **CBD** after decarbossilation
	Benzoic acid	6.5 ± 0.8	0.40 ± 0.1	0.39 ± 0.06	
	TMSP-d_4_	6.4 ± 0.6	0.41 ± 0.1	0.41 ± 0.06	
**CBD** content *	Anthracene	0.4 ± 0.1	4.60 ± 0.8	0.30 ± 0.1	
	Benzoic acid	0.30 ± 0.06	4.54 ± 0.6	2.2 ± 0.1	
	TMSP-d_4_	0.4 ± 0.1	4.59 ± 0.6	2.9 ± 0.1	
**Δ^9^-THC** content *	Anthracene	0.11 ± 0.44	<LOD	<LOD	Regulation (EU) N° 639/2014 [15] 0.09 ± 0.01
	Benzoic acid	0.07 ± 0.02			
	TMSP-d_4_	0.10 ± 0.02			

* % of dry weight.

## Data Availability

All data generated or analyzed during this study are included in this published article.

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
