# Peer review of "NMR Spectroscopy Applied to the Metabolic Analysis of Natural Extracts of Cannabis sativa"

_molecules, 2022, doi:10.3390/molecules27113509_

Round 1
Reviewer 1 Report
Dear author(s):
NMR Spectroscopy Applied to Metabolic Analysis of Natural 2 Extracts of Cannabis sativa
After an exhaustive revision, the manuscript is Reconsider after major revision (control missing in some experiments). In general, the study is closely connected to the journal's objectives. The study is very interesting. The English is good. The introduction is complete, very detailed, but it needs to add references from 2020 to 2022. The section results and discussion should be improved in some points, since the authors must present explication of the results, comparison with other studies, and explication (discussion) of the results obtained with respect to other studies. In the following pages, I give a detailed revision of the manuscript.
ABSTRACT
The abstract is good.
- INTRODUCTION
The introduction is very clear, concise and precise, with good English, and it has updated references until 2019. The authors need to add references from 2020 to 2022.
- MATERIALS AND METHODS
General comments
This section is clear. The English is good. The authors must add a Figure that represents all the complete methodology. This Figure will help to understand the methodology.
2.1. Plant Material and extraction procedure
What is the reference for the extraction methodology?
2.3 NMR samples preparation, experiment, data processing and statistical treatment
What is the reference to NMR sample? What is the reference for the methodology of NMR?
** The authors should make a subsection on statistical analysis **
- RESULTS AND DISCUSSION
The section of “Results and Discussion” is characterized by a very detailed description of the results, explication of the results, comparison with other studies, and explication (discussion) of the results obtained with respect to other studies.
3.1 NMR characterization of seeds extracts
The authors need to eliminate the lines 259-270, since these lines correspond to introduction.
The lines 271-277 correspond to methodology or objective. Thus, the lines must be removed.
The lines 305-313 correspond to methodology. Thus, the lines must be removed.
The authors present a very complete description of the results. However, the explication of the results is weak, i.e., the authors need to add more lines. The authors mention a comparison with other studies, but the authors need to add explication (discussion) of the results obtained with respect to other studies.
3.2 NMR characterization of flower extracts
The authors need to add more description on Figure 4.
This subsection is more complete. However, the authors need to add more comparison with other studies, and explication (discussion) of the results obtained with respect to other studies.
3.3 Multivariate analysis
The authors need to add comparison with other studies, and explication (discussion) of the results obtained with respect to other studies.
3.4 Quantitative analysis of inflorescences
The lines 378-384 correspond to methodology. Thus, the lines must be removed.
The authors need to add comparison with other studies, and explication (discussion) of the results obtained with respect to other studies.
- CONCLUSIONS
The authors should add the section conclusions.
Author Response
Rende 23 May 2022
Dear reviewer,
thank you very much for the effort that you spent in our manuscript. We appreciate the very constructive and helpful comments and suggestions to improve our work. In response to their input, we made the modifications recommended to the text.
We hope that our revised manuscript addresses all concerns satisfactorily.
All changes are presented in details below. The issues raised by reviewer are set in italics and our answers in plain font. All our changes are included in the revised manuscript in red color.
ANSWERS TO REFEREE 1
Open Review
( ) I would not like to sign my review report
(x) I would like to sign my review report
English language and style
( ) Extensive editing of English language and style required
( ) Moderate English changes required
(x) English language and style are fine/minor spell check required
( ) I don't feel qualified to judge about the English language and style
|
Yes |
Can be improved |
Must be improved |
Not applicable |
|
|
Does the introduction provide sufficient background and include all relevant references? |
( ) |
(x) |
( ) |
( ) |
|
Are all the cited references relevant to the research? |
(x) |
( ) |
( ) |
( ) |
|
Is the research design appropriate? |
( ) |
(x) |
( ) |
( ) |
|
Are the methods adequately described? |
( ) |
(x) |
( ) |
( ) |
|
Are the results clearly presented? |
( ) |
( ) |
(x) |
( ) |
|
Are the conclusions supported by the results? |
( ) |
( ) |
(x) |
( ) |
Comments and Suggestions for Authors
Dear author(s):
NMR Spectroscopy Applied to Metabolic Analysis of Natural 2 Extracts of Cannabis sativa
After an exhaustive revision, the manuscript is Reconsider after major revision (control missing in some experiments). In general, the study is closely connected to the journal's objectives. The study is very interesting. The English is good. The introduction is complete, very detailed, but it needs to add references from 2020 to 2022. The section results and discussion should be improved in some points, since the authors must present explication of the results, comparison with other studies, and explication (discussion) of the results obtained with respect to other studies. In the following pages, I give a detailed revision of the manuscript.
Authors: Thank you for the comments and we would like to elucidate each issue raised by the review
ABSTRACT
The abstract is good.
Authors: Thank you
- INTRODUCTION
The introduction is very clear, concise and precise, with good English, and it has updated references until 2019. The authors need to add references from 2020 to 2022.
Authors: We are grateful to the reviewer for noting this as it has given us the opportunity to improve the work by adding references from the last two years.
We added the the following references in the introduction:
“Tura, M.; Ansorena, D.; Astiasarán, I.; Mandrioli, M.; Toschi, T.G. Evaluation of Hemp Seed Oils Stability under Accelerated Storage Test. Antioxidants 2022, 11, 490. https://doi.org/10.3390/antiox 11030490”
“Banskota, A.H.; Jones, A.; Hui, J.P.M.; Stefanova, R. Triacylglycerols and Other Lipids Profiling of Hemp By-Products. Molecules 2022, 27, 2339. https:// doi.org/10.3390/molecules27072339”
“Teleszko, M.; Zaj ˛ac, A.; Rusak, T. Hemp Seeds of the Polish ‘Bialobrzeskie’ and ‘Henola’ Varieties (Cannabis sativa L. var. sativa) as Prospective Plant Sources for Food Production. Molecules 2022, 27, 1448. https://doi.org/10.3390/ molecules270414485”
“M.M. Radwan, S. Chandra, S. Gul, M.A. Elsohly, Cannabinoids, phenolics, terpenes and alkaloids of cannabis, Molecules 26 (2021) 2774. https:// doi.org/10.3390/molecules26092774”
“J.L. Bautista, S. Yu, L. Tian, Flavonoids in Cannabis sativa: biosynthesis, bioactivities, and biotechnology, ACS Omega 6 (2021) 5119e5123. https:// doi.org/10.1021/acsomega.1c00318.”
“Odieka, A.E.; Obuzor, G.U.; Oyedeji, O.O.; Gondwe, M.; Hosu, Y.S.; Oyedeji, A.O. The Medicinal Natural Products of Cannabis sativa Linn.: A Review. Molecules 2022, 27, 1689. https://doi.org/10.3390/ molecules27051689”
“ Iftikhar, A.; Zafar, U.; Ahmed, W.; Shabbir, M.A.; Sameen, A.; Sahar, A.; Bhat, Z.F.; Kowalczewski, P.Ł.; Jarz ˛ebski, M.; Aadil, R.M. Applications of Cannabis Sativa L. in Food and Its Therapeutic Potential: From a Prohibited Drug to a Nutritional Supplement. Molecules 2021, 26, 7699. https://doi.org/10.3390/ molecules26247699”
“Kopustinskiene, D.M.; Masteikova, R.; Lazauskas, R.; Bernatoniene, J. Cannabis sativa L. Bioactive Compounds and Their Protective Role in Oxidative Stress and Inflammation. Antioxidants 2022, 11, 660.”
“Ohtsuki T.; Friesen J.B.; Chen S.; McAlpine J.B.; and Pauli F.G. J. Nat. Prod. 2022, 85, 3, 634–646”
“Stasiłowicz, A.; Tomala, A.; Podolak, I.; Cielecka-Piontek, J. Cannabis sativa L. as a Natural Drug Meeting the Criteria of a Multitarget Approach to Treatment. Int. J. Mol. Sci. 2021, 22, 778. https://doi.org/ 10.3390/ijms22020778”
- MATERIALS AND METHODS
General comments
This section is clear. The English is good. The authors must add a Figure that represents all the complete methodology. This Figure will help to understand the methodology.
Authors: we followed the reviewer's suggestion and we added the following scheme which concisely summarizes the methodology used in the work.
Then, before subsection 2.2 Chemicals and solvents, we added the following scheme, that becomes Figure 2. Consequently, we have also renumbered the other figures.
Figure 2. Schematic experimental steps involved in the extraction, sample preparation and NMR characterization of C. sativa seeds and inflorescences.
2.1. Plant Material and extraction procedure
What is the reference for the extraction methodology?
Authors: Concerning this point, we would like to underline that in the section 2.1. Plant Material and extraction procedure we explicitly said, at lines 152-156 and line 174, that we followed the procedure reported in reference [15] of the original manuscript (Annex III, Commission Delegated Regulation (EU) 2017/1155 of 15 February 2017 Amending Delegated Regulation (EU) No 639/2014 as Regards the Control Measures Relating to the Cultivation of Hemp, Certain Provisions on the Greening Payment, the Payment for Young Farmers in Control of a Legal Person, the Calculation of the per Unit Amount in the Framework of Voluntary Coupled Support, the Fractions of Payment Entitlements and Certain Notification Requirements Relating to the Single Area Payment Scheme and the Voluntary Coupled Support, and Amending Annex X to Regulation (EU) No 1307/2013 of the European Parliament and of the Council. Available online: https://eur lex.europa.eu/eli/reg_del/2017/1155/oj). In that procedure, in the ANNEX III is described the official method for the quantitative determination of THC by gas chromatography (GC) after extraction with a suitable solvent. We use the same protocol for the preparation of NMR samples, of course without adding the standard (squalene) needed for GC analysis. However, to be clearer, in addition to the reference already mentioned, we also include the recent reference https://doi.org/10.1016/j.trac.2022.116554 in which different extraction methods used for C. sativa are reported including Ultrasound-assisted extraction. Moreover, we added also the reference [36] at the end of the section which is related to the extraction procedure for the seeds.
Then, in section 2.1. Plant Material and extraction procedure, at line 173 of original manuscript, after the sentence: “In addition, a quantitative analysis with the gas chromatography (GC) using flame ionization detector (FID) method was conducted on samples of Tiborszallasi variety, prepared from the same dried inflorescence matrix of the NMR samples, by following the protocol reported in literature [15].”
We added:
“It should be emphasized that given the chemical complexity of C. sativa, the extraction and collection of the various bioactive compounds is not simple and for this reason both solvents and different extraction methods are reported in the literature, ranging from microwave assisted extraction to supercritical fluid extraction [36].”
And in the refererences section:
- Liu, Y., Liu; H.-Y.; Li, S.-H.; Ma, W. ; Wu, D.-T.; Li, H.-B.; Xiao, A.-P.; Liu, L.-L.; Zhu, F.; Gan, R.-Y. Cannabis sativa bioactive compounds and their extraction, separation, purification, and identification technologies: An updated review. TrAC. 2022, 149, 116554.
Section 2.1. Plant Material and extraction procedure, at line 180 original manuscript, the sentence “The solution was then paper filtered, evaporated under vacuum at 30°C and the residue was extracted with the same procedure one more time with other 45 mL of same solvent.”
Has been modified as:
“The solution was then paper filtered, evaporated under vacuum at 30°C and the residue was extracted with the same procedure one more time with other 45 mL of same solvent [37]”.
We added also in the references section:
- Rezvankhah, A.; Emam-Djomeh, Z.; Safari, M.; Askari, G.; Salami, M. Investigation on the extraction yield, quality, and thermal properties of hempseed oil during ultrasound-assisted extraction: A comparative study. J. Food Process. Preserv. 2018, e13766.
2.3 NMR samples preparation, experiment, data processing and statistical treatment
What is the reference to NMR sample? What is the reference for the methodology of NMR?
Authors: concerning these points, we would like to underline that the preparation of the NMR samples as well as the use of the common pulse sequences used in the manuscript are routine practices for those working in this field and can be found on the basic texts of the practical NMR. This was the reason for not giving any references for either the samples and the common 1D pulse sequences. Regarding the more sophisticated sequences we have reported in subsection 2.3 various references, such as 2D (references 20,21,22), J-Res (references 23,24), correct calculation of the T1 relaxation time (reference 25). About the methodolgy used in the Introduction section we reported the references 16 and 17 regarding the application of NMR in metabolic analysis of natural matrices. However, to be even clearer we have added same other references related to quantitative analysis.
Then, in the subsection 2.3 NMR samples preparation, experiment and data processing, at line 195 of original manuscript, the sentence: ” For the quantitative analysis, samples of hemp in CDCl3 were prepared carefully weighing all the components and by adding 0.3 mg of internal standard (anthracene, benzoic acid and TMSP-d4). No additional treatment was necessary for the preparation of NMR samples”
Has been replaced by:
“For the quantitative analysis, samples of hemp in CDCl3 were prepared carefully weighing all the components and by adding 0.3 mg of internal standard (anthracene, benzoic acid and TMSP-d4). No additional treatment was necessary for the preparation of NMR samples [39, 40]. “
And we added the following references:
- Hazekamp, A.; Verpoorte, R., Quantitative Analysis of Cannabinoids from Cannabis sativa Using 1H-NMR . Chem. Pharm. Bull. 2004, 52(6), pp. 718-721.
- 40. Marchetti, L; Brighenti, V., Rossi, M.C.; Sperlea, J.; Pellati, F.; Bertelli D., Use of 13C-qNMR Spectroscopy for the Analysis of Non-Psychoactive Cannabinoids in Fibre-Type Cannabis sativa L. (Hemp). Molecules 2019, 24, pp. 1138-1150
** The authors should make a subsection on statistical analysis **
Authors: we followed the reviewer's suggestion and we added in the section 2. MATERIALS AND METHODS the subsection 2.4 Statistical analysis: Principal Component Analysis (PCA) and we modified both the title of subsection 2.3 NMR samples preparation, experiment, data processing and statistical treatment and the last part of the this subsection.
Therefore, the subsection “2.3 NMR samples preparation, experiment, data processing and statistical treatment”
Has been modified as:
“2.3 NMR samples preparation, experiment and data processing”
And we added at the end of this subsection, at line 230 of the original manuscript, the following sentence:
“ For the multivariate statistical analysis, the 1H NMR spectra were segmented in rectangular bucket of fixed 0.05 ppm. The integration region was defined on the first proton spectra and next it was reported, once saved, on the other spectra bucketed automatically. In particular, variables were manually selected by choosing the regions of NMR spectra with characteristic signals of metabolites and eliminating regions with a poor signal-to-noise ratio. Therefore, regions selected for the subsequent statistical treatment were 0.50-1.15 ppm, 1.3 - 2.6 ppm, 3.8-7.0 ppm. Once normalized, the integrals were organized in a data matrix that was mean-centered and scaled. All the data processing steps were carried out using TopSpin 3.6 software (Bruker BioSpin, Rheinstetten, Germany) (TopSpin, 2018) [47].”
At line 230 of the original manuscript we added the new subsection:
“ 2.4 Statistical analysis: Principal Component Analysis (PCA)”
And the following text:
“Multivariate analysis was performed on the 48 1D 1H NMR experiments recorded on the samples prepared from the extracts obtained with different extractive solvents, 24 samples for both varieties, Tiborszallasi, and Kompolti. It should be noted that each sample prepared comes from different extracts, i.e., for example the 9 samples of Kompolti in hexane were obtained from 9 extractive procedures of the corresponding inflorescences. Among the possible statistical approach, in this work the principal component analysis (PCA) was used as explorative method. PCA is a technique able to reduce the dimensionality of the multivariate problem without losing of information. This mathematical treatment allows to clearly visualize samples in a two- or three-dimensional space and this permits to reveal trends in the data or grouping of samples (clusters) based on their similarity [48–50]. This methodology applied to the data matrix of the bucketing 1H NMR spectra of all extract samples, obtained as described in the previous subsection, allowed to obtain two datasets of 24 samples and 130 variables for each variety that were used as starting point to carry out PCA analysis using R software (R software (R Core Team, 2019) [51].”
Then the previous “2.4 Chromatographic experiments” has been modified as:
“2.5 Chromatographic experiments”.
- RESULTS AND DISCUSSION
The section of “Results and Discussion” is characterized by a very detailed description of the results, explication of the results, comparison with other studies, and explication (discussion) of the results obtained with respect to other studies.
3.1 NMR characterization of seeds extracts
The authors need to eliminate the lines 259-270, since these lines correspond to introduction.
Authors: we have deleted these lines from subsection 3.1 NMR characterization of seeds extracts. However we believe that the informations contained in this text are important and therefore we have moved and rearranged them in the Introduction section by adding also more recent references.
Then at Page 1 of the Introduction, line 40 of the original manuscript we modified the sentence:” Cannabis is one of the most ancient and versatile plants sources for intoxicant resin, textile fiber and mostly for seed oil. Hemps seeds are rich in fatty acids with 3:1 ratio ω-6/ω-3 - a very good nutritional value - and for this reason used in the production of functional foods”
As:
“Cannabis is one of the oldest and most versatile sources for the intoxicating resin, for the textile fiber and mostly for phytocannabinoids, extracted from different parts of the plant especially from the inflorescence, and for the seed oil. Hemp seed oil, obtained from Cannabis sativa L. seeds, is highly appreciated for its nutritional, anti-inflammatory, antioxidant and immune-stimulating properties [4]. It is practically free of cannabinoids [5], so it has no psychoactive action, but, like other common vegetable oils, it is rich in essential fatty acids [6]. As reported in several works, this oil is a rich source of ω-3 and ω-6 polyunsaturated fatty acids (almost 80%), in particular linoleic acid (LA) and α-linolenic acid (αLA), with a ratio ω-6 / ω-3 approximately equal to 3:1 [7]. Although various factors, such as cultivation area, cultivar, seed origin, agronomic cultivation practices, etc., affect both the chemical composition and the ω-6 / ω-3 ratio [4,7], this ratio is considered an optimal nutritional value in the prevention of the risk of coronary heart disease [8,9]. Due to this characteristic, Cannabis seed oils are authorized and widely used in the food sector [10], such as the production of functional foods.”
We added also the following references not present in the original manuscript:
“ 5. Tura, M.; Ansorena, D.; Astiasarán, I.; Mandrioli, M.; Toschi, T.G. Evaluation of Hemp Seed Oils Stability under Accelerated Storage Test. Antioxidants 2022, 11, pp. 490-508.”
“6. Banskota, A.H.; Jones, A.; Hui, J.P.M.; Stefanova, R. Triacylglycerols and Other Lipids Profiling of Hemp By-Products. Molecules 2022, 27, 2339. “
“8. Teleszko, M.; Zaj ˛ac, A.; Rusak, T. Hemp Seeds of the Polish ‘Bialobrzeskie’ and ‘Henola’ Varieties (Cannabis sativa L. var. sativa) as Prospective Plant Sources for Food Production. Molecules 2022, 27, 1448.”
The lines 271-277 correspond to methodology or objective. Thus, the lines must be removed.
Authors: we have deleted these lines from subsection 3.1. and we have partly moved them in the Introduction section.
Then, in the Introduction, before the sentence: “ The female inflorescences of the………….”
We added:
“Despite the growing interest in this product, a specific regulation to evaluate its analytical quality parameters is still lacking [7]. In this context, it would be desirable to find methodologies that can provide useful and rapid information both on the chemical composition and on the important ratio w-6/w-3”
The lines 305-313 correspond to methodology. Thus, the lines must be removed.
Authors: We have removed the lines as indicated by the reviewer from subsection 3.1 NMR characterization of seeds extracts, but we moved relations (1) and (2) to subsection 2.3 NMR samples preparation, experiment and data processing.
Then in the subsection 2.3 NMR samples preparation, experiment and data processing, at line 193 of the original manuscript we added:
“Two other similar extraction procedures were repeated on the same starting matrix of dried seeds. On these extracts 1D 1H NMR spectra were recorded to be used for reproducibility and standard deviation in the calculation of essential fatty acids ratio. 1H NMR spectra were manually phased, baseline corrected and the chemical shifts were reported with respect to the TMS signal used as reference. From the 1H NMR spectra of these extracts the main fatty acids ratio w-6/w-3 can be determined by combining the integrals, obtained after applying the deconvolution procedure, of three different signals: (a) the methyl protons of all the acyl groups (LA), with the exception of those of α-linolenic acid; (b) the methyl protons of ω-3 fatty acid (α-linolenic acid (αLA)) and (c) the methylene protons of linoleic and α-linolenic acyl groups; and using the relations [38]: “
|
|
(1) |
and
|
|
(2) |
The authors present a very complete description of the results. However, the explication of the results is weak, i.e., the authors need to add more lines. The authors mention a comparison with other studies, but the authors need to add explication (discussion) of the results obtained with respect to other studies.
Authors: to satisfy the reviewer's request we have modified some parts of subsection: 3.1 NMR characterization of seeds extract.
Then the text at line 301 of original manuscript: “ Three different signals in the protonic spectra were considered: (a) the overlapping triplets at 2.27-2.37 ppm for terminal methyl groups for ω-6 fatty acid (especially linoleic fatty acid (LA)); (b) the triplet at 0.97 ppm generated by the methylenic protons of ω-3 fatty acid (α-linolenic acid; (αLA)); (c) the multiplet at 2.72-2.86 ppm for diallylic protons of linoleic and α-linolenic acids). It’s possible to determine the relative composition in αLA and LA, so the ω-6/ω-3 essential fatty acids ratio using the following equations [35]:”
Has been replaced by:
“Since the integrals of the 1H NMR signals are proportional to the number of hydrogen atoms present in each functional group and, overall, to the number of functional groups present in the sample, from the combination of the integrals of different signals it is possible to calculate the concentration of fatty acids in general and the w-6/w-3 ratio in particular. To this end, three different signals in the protonic spectra were considered: (a) the multiplet at 0.88 ppm due to the overlapping triplet signals of the methyl protons of all the acyl groups (LA), with the exception of those of α-linolenic acid; (b) the triplet at 0.97 ppm generated by the methyl protons of ω-3 fatty acid (α-linolenic acid; (αLA)); (c) the multiplet at 2.72-2.86 ppm generated by diallylic protons of linoleic and α-linolenic acyl groups. By combining the area of these signals, using the relations (1) and (2), that take into account the number of equivalent nuclei in each group, the concentration of αLA and LA was calculated, from which the w-6/w-3 ratio was obtained [38].”
And at line 313, the text: ” As can be seen, this value obtained by NMR is substantially in agreement with the optimal value of the ratio ω-6/ω-3 reported in the literature [31, 32, 36]. This result is doubly important because, on the one hand it indicates the quality of the oily extracts from the seeds of the Futura 75 cultivar grown in Calabria and, on the other hand, it demonstrates that the NMR methodology can measure this parameter in a very rapid way. Indeed, the measurement of this ratio is based only on the recording and analysis of 1H NMR spectra obtained directly from seeds extracts without further derivatization as, instead, is required by GC method, that is the common and validated method used to determine the composition of the oil in terms of fatty acids [37].”
Has been replaced by:
“ As can be seen, this value obtained by 1H NMR is very close to the value of 3:1 for w-6/w-3 considered optimal for humany dietary purpose since it is able to prevent various diseases such as diabetes, cardiovascular disease, cancer, and other chronic disease [4,9,53,54]. On the other hand, the growing interest in hemp seed oil also in other fileds as pharmaceutical and cosmetic [53], leads to a constant search for methods that allow a fast and systematic quality control; the w-6/w-3 ratio is one of these quality parameters [55]. Our result is doubly important because, on the one hand it indicates the quality of the oily extracts from the seeds of the Futura 75 cultivar grown in Calabria and, on the other hand, it confirms that NMR is a reliable quantitative platform for fast screening of hemp oil quality. Indeed, the measurement of this ratio is based only on the recording and analysis of 1H NMR spectra obtained directly from seeds extracts without further derivatization as, instead, is required by Gas Chromatography (GC) method, that is the common and validated method used to determine the composition of the oil in terms of fatty acids [56]. Moreover, our result agrees with what reported in the paper of Siudem et al. [38] in which the authors analysed six different samples of hemp seed oils and for each of them they calculated the ω-6/ω-3 ratio by 1H NMR using the relationships (1) and (2). The authors compared these data with those coming from CG method applied to the same hemp seed oils and found a substantial agreement between them, which proves the effectiveness of the method. It is worth noting that, in a recent paper [53], the 1H NMR methodology as well as being used in the evaluation of the ω-6/ω-3 ratio has been successfully combined with chemometric methods in order to observe the differences in several oil samples due to the different time and storage conditions of the oils. Hence, once again it demonstrates how NMR can be used for rapid and reliable analysis of hemp seed oil quality assessment as an alternative to the more common classical analytical methods.”
And we added new references not present in the original manuscript.
3.2 NMR characterization of flower extracts
The authors need to add more description on Figure 4.
Authors: we have added a more detailed description of experiments 1H and 13C in the legend of the figure 4, of original manuscript, now figure 5. However, we would like to underline that the details, in terms of assignment of the various signals, are shown in the enlargements of figure 7 and therefore we did not consider it necessary to add other details on figure 5.
Then the legend of figure 4, Figure 5 in the revised manuscript: “Figure 4. 500 MHz (a) 1H NMR and (b) 13C-{1H} NMR spectra of inflorescences ethanolic extract dissolved in CDCl3 recorded at 298K.”
Has been replaced by:
“Figure 5. 500 MHz NMR spectra of inflorescence extracts dissolved in CDCl3 recorded at 298K. a) 1H NMR spectrum recorded using zg30 Bruker standard pulses sequence and for each experiment 128 FIDs were accumulated by using a spectral width of 14.00 ppm and a relaxation delay of 5 s. b) 13C-{1H} NMR spectrum (zgig Bruker pulse sequence), performed with proton broad-band decoupling collecting 8K free induction decays (FIDs) using a spectral width 250.00 ppm and a relaxation delay of 5s.1D NMR FIDs were Fourier-transformed, phased, baseline corrected and aligned using the TMS signal as reference. 13C-{1H} NMR spectra were filtered with 1 Hz line broadening before Fourier-transformation.”
This subsection is more complete. However, the authors need to add more comparison with other studies, and explication (discussion) of the results obtained with respect to other studies.
Authors: we have modified some parts of the subsection: 3.2 NMR characterization of flower extracts
At line 338 of the original manuscript, after the references [40-42] we added the following text:
“ It is worth underlining that even in recent papers [35,36,57] the NMR methodology is mostly used to characterize fractions or single components of C. sativa obtained by separation techniques.”
At line 366 of the original manuscript, after the text: “ The assignments for the various cannabinoids present in the ethanolic extract, CBD, CBDA and CBG, of hemp inflorescences…………….”
We added:
“ These results agree with many other literature data [35,39-41] and demonstrate how the non-separative NMR technique, combining 1D and 2D experiments, can be successfully applied directly to complex mixture, such as a Cannabis extract, for chemical characterization. An interesting result of the present study is the measurement of a large number of JH-H, for the main cannabinoids, carried out again directly on the Cannabis extract through the analysis of the 2D J-Res spectra.”
At line 389 of the original manuscript, after the text: “ In particular, this profile is easily recognizable in the samples obtained by using ethanol as extracting solvent. All the signal assignment is reported in the Figure 7.”
We added:
“ As regards the solvents, it must be underline that many are those used for the cannabinoids extraction, but to date there is no a specific protocol. However, it is not surprising that we have better recognized the fatty acid profile in ethanol extracts, since from literature data [58] it seems that ethanol has a greater extraction power than acetone and hexane. On the other hand, hexane, while showing the worst performance in terms of total yield, leads to cleaner extracts, with fewer contaminants, and richer in cannabinoids [59]. For this reason, hexane is used for the selective analysis of cannabinoids in the official method established by the European Commission [29]. It should be noted that this is also evident in our spectra recorded on the hexane extracts which have a cleaner profile than the extracts in the other two solvents and which allow, for the Tiborszallasi variety, to quantify the THC content together with the other cannabinoids.”
3.3 Multivariate analysis
The authors need to add comparison with other studies, and explication (discussion) of the results obtained with respect to other studies.
Authors: to satisfy the reviewer's request we added the following text at the end of subsection 3.3 Multivariate analysis.
At line 442 of original manuscript we added:
“ These results obtained are not surprising since it is well known from the literature [34,60] that the polarity of the solvent affects the chemical composition of the cannabinoids present in the extracts.”
And at line 451 of original manuscript we added:
“ The present study has shown that the exploratory PCA method, although simple, is able to differentiate samples coming from different extraction solvents and, in addition, it allows to highlight also which cannabinoids are the basis of this differentiation. Chemometrics-aided NMR technique on C. sativa components (inflorescences, seed oils, leaves and other less valuable parts of the plant), or on fractions of them, is a widely used methodology to determine different properties of C. sativa as evidenced by the plethora of studies reported in the literature [34,41,61-65 and the references reported therein]. In many of these works the combination of 1H-NMR spectra with chemometric tools was used for the metabolomic differentiation of inflorescence extracts from different cultivars and to identify particular markers in order to discriminate different plant chemotypes [41,61]. In other studies [62,66] targeted and non-targeted NMR methodologies were used to identify and quantify compounds of different classes present in the inflorescence extracts of several C. sativa cultivars and to monitor their variations in three different harvested stages. The cultivars analyzed in these studies have a THC content always below the legal limit while the quantities of the other cannabinoids in the extracts are affected by the harvest time and by the solvent. Although it is not easy to make comparisons given the quantity of factors affecting the composition of cannabinoids, the results obtained in this study substantially agree with the studies described above, and therefore seem to demonstrate the reliability of this method and its possible application to routine analysis of cannabinoids.”
3.4 Quantitative analysis of inflorescences
The lines 378-384 correspond to methodology. Thus, the lines must be removed.
Authors: the lines are the 478-484 and not the 378-384. We moved these line in the subsection 2.3 NMR samples preparation, experiment and data processing, after the sentence: ”Instead, for 13C qNMR quantification experiments (zgig Bruker pulse sequence) were performed collecting 4000 FIDs, using a SW of 250.00 ppm, a relaxation delay of 160.0 s, an acquisition time of 10.0 s.”
The authors need to add comparison with other studies, and explication (discussion) of the results obtained with respect to other studies.
Authors: at line 519, original manuscript, we added the following text:
“It is worth noting that the NMR technique has proven to be a reliable and powerful tool for the quantification of different natural products and, especially in recent years, has also been successfully applied to the quantification of CBD, and other cannabinoids, directly on hemp extracts coming from different cultivars using both the 1H and 13C qNMR methodology. [40, 44, 68-71]. Quantitative 1H-NMR is the most widely used in quantification of natural extracts and has been shown to have a good level of accuracy and reproducibility. However, its application to hemp extracts presents several drawbacks due both to the presence of contaminants and to the overlapping of different signals that require extensive use of deconvolution processes [72]. The use of 13C q-NMR, introduced by Marchetti et al. [40] partially removes these weaknesses and at the same time offers sufficiently precise and sensitive results. As expected, our results substantially agree with these previous reports on the 1H and 13C qNMR investigations even if it is necessary to point out that a direct comparison on the quantities of cannabinoids found in the different extracts of the cultivar analyzed is practically impossible given, as we have already highlighted, the large quantity of variables, i.e. cultivar, geographical origin, harvesting period, agronomic practices, extraction methodologies etc., that affect the composition of cannabinoids. However, the quantitative results of the present study highlight, once again, the remarkable potentialities of the NMR technique which is able to quantify the main metabolites present in the hemp inflorescence extracts we analyzed as they are without further treatment or derivatization as required by the official technique [73,74]. Moreover, as reported recently by Dadiotis et al. [75] concerning the quantitative analysis of cannabinoids in hemp extracts using, in a complementary way, the 1H-NMR and 1H-1H COSY NMR spectra, these quantitative data via NMR are comparable with those acquired with other more consolidated techniques applied to the same extracts. These evidences of a good correspondence between the various quantification techniques are also confirmed by the data we obtained on the Timborszallase cultivar given the satisfactory agreement between the NMR and GC-FID data in hexane solvent.”
- CONCLUSIONS
The authors should add the section conclusions
Authors: Concerning this point raised by the referee, we would like to clarify that in the original manuscript we have already entered the CONCLUSIONS as can be seen at line 532 of the original manuscript, immediately below table 3, where we reported:
- Conclusions
Cannabis sativa is a fast-growing plant currently grown all over the world that is gaining popularity in various fields of research for its biological and pharmaceutical properties. Actually, C. sativa is widely recognized and appreciated for the high nutritional and health-promoting properties of the oil obtained from their seeds together with the pharmacological activity mainly associated to psychoactive and non-psychoactive cannabinoids, chemical components mainly extracted from the inflorescences. In this work, NMR spectroscopy was applied to the analysis of extracts from seeds and inflorescences of different varieties of Cannabis sativa grown in Calabria, in order to explore the potentialities of this technique in the qualitative and quantitative analysis of the extracts and to evaluate the possibility of using it as an alternative to the most common methods in the quantification of cannabinoids present in inflorescence extracts. The quantitative NMR results obtained on two varieties of hemp inflorescence extracts, using different internal standards and solvents, demonstrate the high potentiality of the proposed technique in this field of application. Indeed, the NMR technique has been able to quantify the main cannabinoids present in the extracts; the quantitative data are reproducible; and, most importantly, the data in hexane solvent are congruent with the data obtained by the GC-FID method. Moreover, while this last methodology is not able to distinguish CBD and CBDA, using the NMR method it was possible to separate the two contributions and quantify them. This proved, once again, the analytical power of NMR technique which is able not only to furnish the same results obtained from the official method, including the evaluation of THC, but it also to lead to more informative data without performing particular treatments on the sample.
In addition to the characterization and the quantitative study, different extraction procedures were tested and evaluated by NMR spectroscopy with the aim to obtain inflorescence extracts poor in psychotropic agents and rich in medical cannabinoids and triacylglicerols (TAGs) whose w-6/w-3 ratio has been found to be excellent from a nutritional point of view. Specifically, extracts of inflorescences obtained by ultrasound-assisted solute-solvent extraction using hexane, acetone and ethanol as solvents, were studied. By elaborating the spectral data with a statistical method (PCA) together with the qNMR approach, it was possible to conclude that hexane is more efficient in the extraction of cannabinoids (THC included) than the TAGs constituents while extraction with a higher polarity solvent (acetone or ethanol) allows to obtain samples free from THC (THC content < LOD), with a lower percentage of cannabinoids and richer in TAGs. This evidence can be exploited to obtain extracts rich in bioactive compounds, both cannabinoids and TAGs, potentially usable in the food and pharmaceutical industry, opening new paths in the production of functional foods and supplements.
Authors: since we have substantially changed the References section, adding many more references than those present in the original manuscript, we report it in full below.
References
- Small, E. Classification of Cannabis Sativa in Relation to Agricultural, Biotechnological, Medical and Recreational Utilization. In Cannabis Sativa L.- Botany and Biotechnology, 1st ed.; Chandra, S., Hemant, L., ElSohly, M.A., Eds.; Springer International Publishing: Cham, Switzerland, 2017; pp. 1-62.
- Li, H.L. An archaeological and historical account of Cannabis in China. Bot. 1974, 28(4), pp. 437-447.
- ElSolhy, M.A.; Tadwan, M.M.; Gul, W.; Chandra, S.; Galal, A. Phytochemistry of Cannabis sativa L.. Chem. Org. Nat. Prod. 2017, 103, pp. 1-36.
- Crescente, G.; Piccolella, S.; Esposito, A.; Scognamiglio, M.; Fiorentino, A.; Pacifico, S. Chemical composition and nutraceutical properties of hempseeds: an ancient food with actual functional value. Rev. 2018, 17, pp. 733-749.
- Tura, M.; Ansorena, D.; Astiasarán, I.; Mandrioli, M.; Toschi, T.G. Evaluation of Hemp Seed Oils Stability under Accelerated Storage Test. Antioxidants 2022, 11, pp. 490-508.
- Banskota, A.H.; Jones, A.; Hui, J.P.M.; Stefanova, R. Triacylglycerols and Other Lipids Profiling of Hemp By-Products. Molecules 2022, 27, 2339.
- Spano, M.; Di Matteo, G.; Rapa, M.; Ciano, S.; Ingallina, C.; Cesa, S.; Menghini, L.; Carradori, S.; Giusti, A.M.; Di Sotto, A.; Di Giacomo, S.; Sobolev, A.P.; Vinci, G.; Mannina, L. Commercial Hemp Seed Oils: A Multimethodological Characterization. Sci., 2020, 10(19), 6933.
- Teleszko, M.; Zaj ˛ac, A.; Rusak, T. Hemp Seeds of the Polish ‘Bialobrzeskie’ and ‘Henola’ Varieties (Cannabis sativa L. var. sativa) as Prospective Plant Sources for Food Production. Molecules 2022, 27, 1448.
- Simopoulos, A.P. The importance of the omega-6/omega-3 fatty acid ratio in cardiovascular disease and other chronic disease. Biol. Med. 2008, 233(6), pp. 674-688.
- Ministero della Salute. Produzione e Commercializzazione di Prodotti a Base di Semi di Canapa Per L’utilizzo nci Settori Dell’alimentazione Umana; Ministero della Salute: Roma, Italy, 2009; pp. 1–4.
- Mercuri, A.M.; Accorsi, C.A.; Bandini Mazzanti, M. The long history of Cannabis and its cultivation by Romans in central Italy, shown by pollen records from Lago Albano and Lago di Nemi. Hist. Archaeobot. 2002, 11, pp. 263-276.
- ElSolhy, M.A.; Slade, D. Chemical Constituents of marijuana: the complex mixture of natural cannabinoids. Life 2005, 78(5), pp. 539-548.
- Radwan, M.M, Chandra, S., Gul, S., Elsohly, M.A. Cannabinoids, phenolics, terpenes and alkaloids of cannabis, Molecules 2021, 26, 2774.
- Bautista, J.L., Yu, S., Tian, L. Flavonoids in Cannabis sativa: biosynthesis, bioactivities, and biotechnology, ACS Omega 2021, 6, 5119-5123.
- Baker, D.; Pryce, G.; Giovannoni, G.; Thompson, A.J. The therapeutic potential of Cannabis. Neurol. 2003, 2(5), pp. 191-298.
- Izzo, A.A.; Borrelli, F.; Capasso, R.; Di Marzo, V.; Mechoulam, R. Non-psychotropic plant cannabinoids: new therapeutic opportunities from an ancient herb. Trends Pharmacol. Sci. 2009, 30(10), pp. 515-527.
- Odieka, A.E.; Obuzor, G.U.; Oyedeji, O.O.; Gondwe, M.; Hosu, Y.S.; Oyedeji, A.O. The Medicinal Natural Products of Cannabis sativa Linn.: A Review. Molecules 2022, 27, 1689.
- Iftikhar, A.; Zafar, U.; Ahmed, W.; Shabbir, M.A.; Sameen, A.; Sahar, A.; Bhat, Z.F.; Kowalczewski, P.Ł.; Jarz ˛ebski, M.; Aadil, R.M. Applications of Cannabis Sativa L. in Food and Its Therapeutic Potential: From a Prohibited Drug to a Nutritional Supplement. Molecules 2021, 26, 7699.
- Russo, B. Cannabis Therapeutics and the Future of Neurology. Front. Integr. Neurosci. 2018, 12:51.
- Stasiłowicz, A.; Tomala, A.; Podolak, I.; Cielecka-Piontek, J. Cannabis sativa L. as a Natural Drug Meeting the Criteria of a Multitarget Approach to Treatment. J. Mol. Sci. 2021, 22, 778.
- Kopustinskiene, D.M.; Masteikova, R.; Lazauskas, R.; Bernatoniene, J. Cannabis sativa L. Bioactive Compounds and Their Protective Role in Oxidative Stress and Inflammation. Antioxidants 2022, 11, 660.
- Pisanti, S.; Bifulco, M. Medical Cannabis: A plurimillennial history of an evergreen. Cell. Physiol. 2019, 234(6), pp. 8342-8351.
- European Commission (2009) Regulation (EC) No 1107/2009 of the European Parliament and of the Council of 21 October 2009 concerning the placing of plant protection products on the market and repealing Council Directives 79/117/EEC and 91/414/EEC. J. Eur. Union 2009, L 309/1.Avaible online: https://eur-lex.europa.eu/legal-content/EN/TXT/?uri=celex%3A32009R1107 (accessed on 14 April 2022).
- Schachtsiek, J.; Warzecha, H.; Kayser, O.; Stehle, F. Current Perspectives on Biotechnological Cannabinoid Production in Plants. Planta Med. 2018, 84(4), pp. 214-220.
- Brunetti, P.; Pichini, S.; Pacifici, R.; Busardò, F.P.; del Rio, A. Herbal preparations of medical cannabis: A vademecum for prescribing doctors. Medicina 2020, 56, 237.
- Legge 2 Dicembre 2016, n.242. Disposizioni per la Promozione della Coltivazione e della Filiera Agroindustriale della Canapa (16G00258), GU Serie Generale n. 304 del 30-12-2016. Available online: https://www.gazzettaufficiale.it/eli/id/2016/12/30/16G00258/sg (accessed on 20 April 2022).
- Circolare 31 Luglio 2018, prot. 2018/43586. Aspetti giuridico-operativi connessi al fenomeno della commercializzazione delle infiorescenze della canapa tessile a basso tenore di THC e relazioni con la normativa sugli stupefacenti. Available online: https://www.camera.it/temiap/2018/12/14/OCD177-3851.pdf (accessed on 02 April 2022).
- Decreto 4 novembre 2019 Definizione di livelli massimi di tetraidrocannabinolo (THC) negli alimenti. (20A00016) (GU Serie Generale n.11 del 15-01-2020). Available online: https://www.gazzettaufficiale.it/eli/id/2020/01/15/20A00016/sg (accessed on 20 Aprile 2022).
- Annex III, Commission Delegated Regulation (EU) 2017/1155 of 15 February 2017 Amending Delegated Regulation (EU) No 639/2014 as Regards the Control Measures Relating to the Cultivation of Hemp, Certain Provisions on the Greening Payment, the Payment for Young Farmers in Control of a Legal Person, the Calculation of the per Unit Amount in the Framework of Voluntary Coupled Support, the Fractions of Payment Entitlements and Certain Notification Requirements Relating to the Single Area Payment Scheme and the Voluntary Coupled Support, and Amending Annex X to Regulation (EU) No 1307/2013 of the European Parliament and of the Council. Available online: https://eur-lex.europa.eu/eli/reg_del/2017/1155/oj (accessed on 02 April 2022 ).
- Vignoli, A.; Ghini, V.; Meoni, G.; Licari, C.; Takis, P.G.; Tenori, L.; Turano, P.; Luchinat, C. High-Throughput Metabolomics by 1D NMR. Chem. Int. 2019, 58(4), pp. 968-994.
- Eisenmann, P.; Ehlers, M.; Weinert, C.H.; Tzvetkova, P.; Silber, M.; Rist, M.J.; Luy, B.; Muhle-Goll, C. Untargeted NMR Spectroscopic Analysis of the Metabolic Variety of New Apple Cultivars. Metabolites 2016, 6(3):29.
- Chandra, S.; Lata, H.; Khan, I.A.; ElSohly, M.A. Cannabis Sativa - Botany and Horticulture. In Cannabis Sativa L.- Botany and Biotechnology, 1st ed.; Chandra, S., Hemant, L., ElSohly, M.A., Eds.; Springer International Publishing: Cham, Switzerland, 2017; pp. 79-100.
- Aizpurua-Olaizola, O.; Soydaner, U.; Öztürk, E.; Schibano, D.; Simsir, Y.; Navarro, P.; Etxebarria, N.; Usobiaga, A. Evolution of the Cannabinoid and Terpene Content during the Growth of Cannabis sativa Plants from Different Chemotypes. Nat. Prod. 2016, 79(2), pp. 324-331.
- Brighenti, V., Protti, M., Anceschi, L., Zanardi, C., Mercolini, L., Pellati, F. Emerging challenges in the extraction, analysis and bioanalysis of cannabidiol and related compounds. J Pharm Biomed Anal. 2021, 192, 113633.
- Ohtsuki, T., Friesen, J.B., Chen, S.N., McAlpine, J.B., Pauli, G.F. Selective Preparation and High Dynamic-Range Analysis of Cannabinoids in "CBD Oil" and Other Cannabis sativa Preparations. Nat. Prod. 2022, 85(3), pp. 634-646.
- Liu, Y., Liu; H.-Y.; Li, S.-H.; Ma, W. ; Wu, D.-T.; Li, H.-B.; Xiao, A.-P.; Liu, L.-L.; Zhu, F.; Gan, R.-Y. Cannabis sativa bioactive compounds and their extraction, separation, purification, and identification technologies: An updated review. TrAC. 2022, 149, 116554.
- Rezvankhah, A.; Emam-Djomeh, Z.; Safari, M.; Askari, G.; Salami, M. Investigation on the extraction yield, quality, and thermal properties of hempseed oil during ultrasound-assisted extraction: A comparative study. Food Process. Preserv. 2018, e13766.
- Siudem, P.; Wawer, I.; Paradowska, K. Rapid evaluation of edible hemp oil quality using NMR and FT-IR spectroscopy. Mol. Struct. 2019, 1177, pp. 204-208.
- Hazekamp, A.; Choi, Y.H.; Verpoorte, R. Quantitative Analysis of Cannabinoids from Cannabis sativa using 1H- NMR. Pharm. Bull. 2004, 52(6), pp. 718-721.
- Marchetti, L.; Brighenti, V.; Rossi, M.C.; Sperlea, J.; Pellati, F.; Bertelli, D. Use of 13C-qNMR Spectroscopy for the Analysis of Non-Psychoactive Cannabinoids in Fibre-Type Cannabis sativa L. (Hemp). 2019, 24(56):1138.
- Choi, Y.H.; Kim, H.K.; Hazekamp, A.; Erkelens, C.; Lefeber, A.W.M.; Verpoorte, R. Metabolomic differentiation of Cannabis sativa cultivars using 1H NMR spectroscopy and principal component analysis. Nat. Prod. 2004, 67(6), pp. 953-957.
- Ludwig, C.; Viant, M.R. Two-dimensional J-resolved NMR spectroscopy: review of a key methodology in the metabolomics toolbox. Anal. 2010, 21(1), pp. 22-32.
- Huang, Y.; Zhang, Z.; Chen, H.; Feng, J.; Cai, S.; Chen, Z. A high-resolution 2D J-resolved NMR detection technique for metabolite analyses of biological samples. Rep. 2015, 5:8390.
- Brighenti, V.; Marchetti, L.; Anceschi, L.; Protti, M.; Verri, P.; Pollastro, F.; Mercolini, L.; Bertelli, D.; Zanardi, C.; Pellati, F. Separation and non-separation methods for the analysis of cannabinoids in Cannabis sativa L.. Pharm. Biomed. 2021, 206:114346.
- Kumar Bharti, S.; Roy, R. Quantitative 1H NMR Spectroscopy. 2012, 35, pp. 5-26.
- Araneda, J.F.; Chu, T.; Leclerc, M.C.; Riegel, S.D.; Spingarn, N. Quantitative analysis of cannabinoids using benchtop NMR instruments. Methods 2020, 12, pp. 4853-4857.
- TopSpin (2018). Avaible on: http://www.bruker-biospin.com/topspin.html
- Varmuz, ; Filzmoser, P. Introduction to Multivariate Statistical Analysis in Chemometrics, 1st ed.; CRC Press Taylor & Francis Group: Boca Raton, Florida, 2009, pp.59-101.
- Ebrahimi, P.; Viereck, N.; Bro, R.; Engelsen, S.B. Chemometric Analysis of NMR Spectra. In Modern Magnetic Resonance, 2nd ed.; Webb, A.G., Ed.; Springer International Publishing: Cham, Switzerland, 2018; pp. 1649-1668.
- Ren, S.; Hinzman, A.A.; Kang, E.L.; Szczesniak, R.; Lu, L.J. Computational and Statistical Analysis of Metabolomics Data. Metabolomics 2015, 11(6), pp.1492-1513.
- R Core Team (2019). R: A language and environment for statistical computing. R Foundation for Statistical Computing, Vienna, Austia. Avaible on: https://www.R-project.org/.
- Popescu, R.; Costinel, D.; Dinca, O.R.; Marinescu, A.; Stefanescu, I.; Ionete R.E. Discrimination of vegetable oils using NMR spectroscopy and chemometrics. Food Control 2015, 48, pp. 84-90.
- Siudem, P.; Wawer, I.; Kowalska, V.; Paradowska, K. Rapid 1H NMR and chemometric methods in verification of hemp-seed oil quality. Pharm. Biomed. 2022, 212: 114650.
- Mikulcová, V.; Kašpárková, V.; Humpolíˇcek, P.; Bunková, L. Formulation, Characterization and Properties of Hemp Seed Oil and Its Emulsions. Molecules 2017, 22, 700.
- Farinon, B.; Costantini, L.; Molinari, R.; Di Matteo, G.; Garzoli, S.; Ferri, S.; Ceccantoni, B.; Mannina, L.; Merendino, N. Effect of malting on nutritional and antioxidant properties of the seeds of two industrial hemp (Cannabis sativa ) cultivars. Food Chemistry 2022, 370, 131348.
- Borges, G.R.; Birk, L.; Scheid, C.; Morés, L.; Carasek, E.; Kitamura, R.O.S.; Roveri, F.; Eller, S.; de Oliveira Merib, J.; de Oliveria, T.F. Simple and straightforward analysis of cannabinoids in medicinal products by fast-GC–FID. Forensic Toxicol. 2020, 38, pp. 531-535.
- Mazzara, E.; Torresi, J.; Fico, G.; Papini, A.; Kulbaka, N.; Dall’Acqua, S.; Sut, S.; Garzoli, S.; Mustafa, A.M.; Cappellacci, L.; et al. A Comprehensive Phytochemical Analysis of Terpenes, Polyphenols and Cannabinoids, and Micromorphological Characterization of 9 Commercial Varieties of Cannabis sativa Plants 2022, 11, 891
- Brighenti, V.; Pellati, F.; Steinbach, M.; Maran, D.; Benvenuti, S. Development of a new extraction technique and HPLC method for the analysis of non-psychoactive cannabinoids in Cannabis sativa L. fibosa (hemp). Pharm. Biomed. Anal. 2017; 143: 228–236.
- Pellati, F.; Brighenti,V.; Sperlea, J.; Marchetti, L.; Bertelli, D.; Benvenuti, S. New methods for the comprehensive analysis of bioactive compounds in Cannabis sativa L. (hemp). Molecules. 2018; 23 (10), 2639.
- Valizadehderakhshan, M.; Shahbazi, A.; Kazem-Rostami, M.; Todd, M.S.; Bhowmik, A.; Wang, L. Extraction of Cannabinoids from Cannabis sativa (Hemp)—Review. Agriculture 2021, 11, 384.
- Peschel, W.; Politi, M. 1H NMR and HPLC/DAD for Cannabis sativaChemotype distinction, extract profiling and specification, Talanta 2015, 140, pp. 150–165.;
- Spano, M.; Di Matteo, G.; Ingallina, C.; Botta, B.; Quaglio, D.; Ghirga, F.; Balducci, S.; Cammarone, S.; Campiglia, E.; Giusti, A.M.; et al. Emerging challenges in the extraction, analysis and bioanalysis of cannabidiol and related compounds. Molecules 2021, 26, 2912.
- Monti, M.C.; Frei, P.; Weber, S.; Scheurer, E.;· Mercer‑Chalmers‑Bender, K. Beyond Δ9‑tetrahydrocannabinol and cannabidiol: chemical differentiation of cannabis varieties applying targeted and untargeted analysis. Bioanal. Chem., 2022, 414, pp. 3847–3862.
- Palmieri S.; Mascini, M.; Ricci, A.; Fanti F.; Ottaviani C.; Lo Sterzo, C.; Sergi, M. Identification of Cannabis sativa L. (hemp) Retailers by Means of Multivariate Analysis of Cannabinoids. Molecules 2019, 24, 3602.
- Mudge, E.M.; Murch, S.J.; Brown, P.N. Chemometric Analysis of Cannabinoids: Chemotaxonomy and Domestication Syndrome. Rep. 2018, 8, 13090.
- Ingallina, C.; Sobolev,A.P; Circi, S.; Spano, M.; Fraschetti,C.; Filippi, A.; et al. Cannabis sativa L. Inflorescences from Monoecious Cultivars Grown in Central Italy: An Untargeted Chemical Characterization from Early Flowering to Ripening. Molecules 2020, 25, 1908.
- Olasehinde, T.A.; Olaniran, A.O. Neurotoxicity of anthracene and benz[a]anthracene involves oxidative stress-induced neuronal damage, cholinergic dysfunction and disruption of monoaminergic and purinergic enzymes. Res. 2022, 80, 105312.
- Nagy, D.U; Cianfaglione, K.; Maggi, F.; Sut, S.; Dall’Acqua, S. Chemical Characterization of Leaves, Male and Female Flowers from Spontaneous Cannabis (Cannabis sativa L.) Growing in Hungary. Biodiversity. 2019, 16, e1800562.
- Kostas Ioannidis, K.; Dadiotis, E.; Mitsis, V.; Melliou, E.; Magiatis, P. Biotechnological Approaches on Two High CBD and CBG Cannabis sativa L. (Cannabaceae) Varieties: In Vitro Regeneration and Phytochemical Consistency Evaluation of Micropropagated Plants Using Quantitative 1H-NMR. Molecules 2020, 25, 5928.
- Siciliano, C.; Bartella, L.; Mazzotti, F.; Aiello, D.; Napoli A.; De Luca, P.; Temperini, A. 1H NMR quantification of cannabidiol (CBD) in industrial products derived from Cannabis sativa L. (hemp) seeds. IOP Conf. Ser.: Mater. Sci. Eng. 2019,572, 012010.
- Barthlott, I.; Scharinger, A.; Golombek, P.; Kuballa, T.; Lachenmeier, D.W. A Quantitative 1H NMR Method for Screening Cannabinoids in CBD Oils. Toxics 2021, 9, 136.
- Risoluti, R.; Gullifa, G.; Battistini, A.; Materazzi, S. Monitoring of cannabinoids in hemp flours by MicroNIR/Chemometrics. Talanta. 2020, 211:120672.
- Sgrò, S.; Lavezzi, B.; Caprari, C.; Polito, M.; D’Elia, M.; Lago, G.; Furlan, G.; Girotti, S.; Ferri, E.N. Delta9-THC determination by the EU official method: evaluation of measurement uncertainty and compliance assessment of hemp samples. Bioanal. Chem. 2021, 413, 3399-3410.
- Nahar, L.; Guo, M.; Sarker, S.D. Gas chromatographic analysis of naturally occurring cannabinoids: A review of literature published during the past decade. Anal. 2020, 31(2), pp. 135-146.
- Dadiotis, E.; Mitsis, V.; Melliou, E.; Magiatis, P. Direct Quantitation of Phytocannabinoids by One-Dimensional 1H qNMR and Two-Dimensional 1H-1H COSY qNMR in Complex Natural Mixtures. Molecules 2022, 27, 2965.

Reviewer 2 Report
The manuscript entitled “NMR Spectroscopy Applied to Metabolic Analysis of Natural Extracts of Cannabis sativa” describes the application of NMR spectroscopy to provide qualitative and quantitative information on the cannabinoids and fatty acids composition of seed and flower extracts from C. sativa.
The study rationale, procedures and methods are generally appropriate, but needs certain modification before it is accepted for publication in the Journal.
I have the following comments:
- In the introduction, more references are needed about the fatty acids composition (lines 40-43).
- About the identification and quantification of unsaturated fatty acids in seed extracts is not clear how the authors can discriminate the presence of saturated fatty acids in the mixture. In particular, the paragraph at lines 300-305 should be revised because the NMR assignment is wrong. Moreover, they follow the two methyl signals at 0.97ppm and 0.88 ppm to discriminate respectively, ω3 and ω6 fatty acids. The typical signal of the terminal methyl of saturated fatty acids is supposed to be overlapped with ω6 methyl (δH 0.88). How the authors comment that? How they considered that in the quantitative analysis?
- On the other hand, the phytochemical characterization of the two cannabis varieties Tiborszallasi, and Kompolti by NMR resulted in the identification of THC in the Tiborszallasi hexane extracts and the totally undetectable presence in any Kompolti These information are not consistent with the experimental data reported in the supporting information. Please, use the same spectral width in the 1H NMR spectra in the figure S1-S2.
- I would suggest to use the cannabinoid numbering systems used in the recent review Prod. Rep., 2016, 33, 1357–1392.
- In the NMR assignment discussion I would suggest to use the notation H-9 instead H9. Normally the subscript indicates the number of atoms.
The English is rough in this manuscript but it's suitable for review purposes. The language issues can be dealt with in review.
Author Response
Rende 23 May 2022
Dear reviewer,
thank you very much for the effort that you spent in our manuscript. We appreciate the very constructive and helpful comments and suggestions to improve our work. In response to their input, we made the modifications recommended to the text.
We hope that our revised manuscript addresses all concerns satisfactorily.
All changes are presented in details below. The issues raised by the reviewer are set in italics and our answers in plain font. All our changes are included in the revised manuscript in red color.
Comments and Suggestions for Authors
The manuscript entitled “NMR Spectroscopy Applied to Metabolic Analysis of Natural Extracts of Cannabis sativa” describes the application of NMR spectroscopy to provide qualitative and quantitative information on the cannabinoids and fatty acids composition of seed and flower extracts from C. sativa.
The study rationale, procedures and methods are generally appropriate, but needs certain modification before it is accepted for publication in the Journal.
Authors: Thank you for your comments and below are our responses to the various comments
I have the following comments:
- In the introduction, more references are needed about the fatty acids composition (lines 40-43).
Authors: Concerning this point, we would like to point out that in the Results and Discussion section, subsection 3.1 NMR characterization of seeds extracts, we have reported some references, in particular 31, 32 and 33 ( in the original manuscript), concerning the composition and the properties related to the w-6 / w-3 ratio which is generally found in hemp seed oil. However, following the referee's advices, we anticipated these references in the Introduction where we also add other references and, at the same time, we have also modified the text (lines 40-43 of the Introduction) by moving and modifying much of the initial text of the subsection 3.1 NMR characterization of seed extracts (lines 259 to 270) in the Introduction section.
Then at Page 1 of the Introduction, the sentence: “ Cannabis is one of the most ancient and versatile plants sources for intoxicant resin, textile fiber and mostly for seed oil. Hemps seeds are rich in fatty acids with 3:1 ratio ω-6/ω-3 - a very good nutritional value - and for this reason used in the production of functional foods.”
Has been modified as:
“ Cannabis is one of the oldest and most versatile sources for the intoxicating resin, for the textile fiber and mostly for phytocannabinoids, extracted from different parts of the plant especially from the inflorescence, and for the seed oil. Hemp seed oil, obtained from Cannabis sativa L. seeds, is highly appreciated for its nutritional, anti-inflammatory, antioxidant and immunostimulating properties [4]. It is practically free of cannabinoids [5], so it has no psychoactive action, but, like other common vegetable oils, it is rich in essential fatty acids [6]. As reported in several works, this oil is a rich source of ω-3 and ω-6 polyunsaturated fatty acids (almost 80%), in particular linoleic acid (LA) and α-linolenic acid (αLA), with a ratio ω-6 / ω-3 approximately equal to 3:1 [7]. Although various factors, such as cultivation area, cultivar, seed origin, agronomic cultivation practices, etc., affect both the chemical composition and the ω-6 / ω-3 ratio [4,7], this ratio is considered an optimal nutritional value in the prevention of the risk of coronary heart disease [8, 9]. Due to this characteristic, Cannabis seed oils are authorized and widely used in the food sector [10], such as the production of functional foods.”
We added also the following references not present in the original manuscript:
“ 5. Tura, M.; Ansorena, D.; Astiasarán, I.; Mandrioli, M.; Toschi, T.G. Evaluation of Hemp Seed Oils Stability under Accelerated Storage Test. Antioxidants 2022, 11, pp. 490-508.”
“6. Banskota, A.H.; Jones, A.; Hui, J.P.M.; Stefanova, R. Triacylglycerols and Other Lipids Profiling of Hemp By-Products. Molecules 2022, 27, 2339. “
“8. Teleszko, M.; Zaj ˛ac, A.; Rusak, T. Hemp Seeds of the Polish ‘Bialobrzeskie’ and ‘Henola’ Varieties (Cannabis sativa L. var. sativa) as Prospective Plant Sources for Food Production. Molecules 2022, 27, 1448.”
- About the identification and quantification of unsaturated fatty acids in seed extracts is not clear how the authors can discriminate the presence of saturated fatty acids in the mixture. In particular, the paragraph at lines 300-305 should be revised because the NMR assignment is wrong. Moreover, they follow the two methyl signals at 0.97ppm and 0.88 ppm to discriminate respectively, ω3 and ω6 fatty acids. The typical signal of the terminal methyl of saturated fatty acids is supposed to be overlapped with ω6 methyl (δH 0.88). How the authors comment that? How they considered that in the quantitative analysis?
Authors: thank’s to the referee for the comment and for notice that. Actually the referee's comment is appropriate because the assignments have been reported incorrectly by us in the text, but the w-6/w-3 ratio calculations are correct. Therefore: (a) the signal centered at 0.88 ppm is a multiplet due to the overlapping of the triplet signals of the methyl protons of all the acyl groups, with the exception of those of α-linolenic acid; (b) the signal of triplet centered at 0.97 ppm, which is easily recognized in the spectrum, is assigned to the methyl protons of α-linolenic acid; (c) the signal of multiplet at 2.72 -2.86 ppm is due to the diallylic protons of linoleic and α-linolenic acids. The integrals of these three signals, indicated with (a), (b) and (c) respectively and obtained by applying the deconvolution procedure for each of them, have been used in the equations (1) and (2) of the manuscript to calculate the ratio w-6/w-3. It should be noted that these equations have already been applied in other works as reported in reference [35] of the original manuscript ( Paweł. Siudem, I. Wawer, K. Paradowska, Rapid evaluation of edible hemp oil quality using NMR and FT-IR spectroscopy, Journal of Molecular Structure (2018), doi: https://doi.org/10.1016/j.molstruc.2018.09.057) and in the references contained therein. In particular, in this reference, the authors took into consideration six different hemp seed oils and for each of them they calculated the w-6/w-3 ratio using the equations (1) and (2). The authors compared these data with those coming from the GC method applied to the same hemp oils and they found a good agreement between them, which demonstrates the validity of the NMR methodology.
According to the indication of the referee we have revised the assignments as follows:
Subsection 3.1 NMR characterization of seeds extracts, line 301- 305 of original manuscript, the sentence: “ Three different signals in the protonic spectra were considered: (a) the overlapping triplets at 2.27-2.37 ppm for terminal methyl groups for ω-6 fatty acid (especially linoleic fatty acid (LA)); (b) the triplet at 0.97 ppm generated by the methylenic protons of ω-3 fatty acid (α-linolenic acid; (αLA)); (c) the multiplet at 2.72-2.86 ppm for diallylic protons of linoleic and α-linolenic acids).”
Has been replaced by:
“Since the integrals of the 1H NMR signals are proportional to the number of hydrogen atoms present in each functional group and, overall, to the number of functional groups present in the sample, from the combination of the integrals of different signals it is possible to calculate the concentration of fatty acids in general and the w-6/w-3 ratio in particular. To this end, three different signals in the protonic spectra were considered: (a) the multiplet at 0.88 ppm due to the overlapping triplet signals of the methyl protons of all the acyl groups (LA), with the exception of those of α-linolenic acid; (b) the triplet at 0.97 ppm generated by the methyl protons of ω-3 fatty acid (α-linolenic acid; (αLA)); (c) the multiplet at 2.72-2.86 ppm generated by diallylic protons of linoleic and α-linolenic acyl groups. By combining the area of these signals, using the relations (1) and (2), that take into account the number of equivalent nuclei in each group, the concentration of αLA and LA was calculated, from which the w-6/w-3 ratio was obtained [38].”
Moreover, we have corrected the assignments in Table 1 as well. Then table 1:
Table 1 : Table 1. 1H NMR chemical shifts and 1H/1H correlations of fatty acids protons in triacylglicerols (TAGs) in CDCl3 for hemp seeds extracts.
|
position |
δH, multiplicitya (J in Hz) |
COSY |
|
A |
2.27-2.37, m |
E |
|
B |
1.55-1.67, m |
C, A |
|
C |
1.23-1.39, m |
Gω6 B, D |
|
D |
1.98-2.11, m |
Gω3, C, F, E |
|
E |
5.28-5.42, m |
D, F |
|
F |
2.76, t |
D, E |
|
Gω3 |
0.97, t |
D |
|
Gω6 |
0.88, t |
C |
|
H, L (Glya) |
4.14, dd (11.88, 5.93) |
H’, L’, I |
|
I (Glya) |
5.26, m |
H, H’, L, L’ |
|
H’, L’ (Glya) |
4.29, dd (11.88, 4.31) |
H, L, I |
aAbbreviations: s, singlet; d, doublet; dd, doublet of doublet; t, triplet; m, multiplet; Gly, Glycerol
has been replaced by:
Table 1. 1H NMR chemical shifts and 1H/1H correlations of fatty acids protons in triacylglicerols (TAGs) in CDCl3 for hemp seeds extracts.
|
position |
δH, multiplicitya (J in Hz) |
COSY |
|
A |
2.27-2.37, m |
E |
|
B |
1.55-1.67, m |
C, A |
|
C |
1.23-1.39, m |
Gω6 B, D |
|
D |
1.98-2.11, m |
Gω3, C, F, E |
|
E |
5.28-5.42, m |
D, F |
|
F |
2.72-2.86, m |
D, E |
|
Gω3 |
0.97, t |
D |
|
Gω6 |
0.88, m |
C |
|
H, L (Glya) |
4.14, dd (11.88, 5.93) |
H’, L’, I |
|
I (Glya) |
5.26, m |
H, H’, L, L’ |
|
H’, L’ (Glya) |
4.29, dd (11.88, 4.31) |
H, L, I |
aAbbreviations: dd, doublet of doublet; t, triplet; m, multiplet; Gly, Glycerol
- On the other hand, the phytochemical characterization of the two cannabis varieties Tiborszallasi, and Kompolti by NMR resulted in the identification of THC in the Tiborszallasi hexane extracts and the totally undetectable presence in any Kompolti These information are not consistent with the experimental data reported in the supporting information. Please, use the same spectral width in the 1H NMR spectra in the figure S1-S2.
Authors: Concerning this point, we reiterated in several parts of the manuscript (subsections 3.2 NMR characterization of flower extracts and 3.4 Quantitative analysis of inflorescences) that only for the hexane extracts of Tiborszallasi we can detect the signal of the THC in the NMR spectra and we reported our qNMR data in the Table 3 and Table S1 for the two varieties. However it is not easy to show every detail on the spectra and we understand that the figures S1 and S2 shown in supporting informations are not very clear, so we have tried to improve them and, following the advice of the referee, we use the same spectral width. Moreover, to better highlight the differences between the two cultivars we have added figure S3 in which we compare the two extracts in hexane and in the enlargement we show the area around 6.4 ppm where in Tiborszallasi spectrum the peak of THC is clearly visible while in Kompolti it is missing.
The new figure are:
Figure S1 Comparison between 1H NMR spectra of ethanol (blue), acetone (red) and hexane (green) extracts for Tiborszallasi variety.
Figure S2 Comparison between 1H NMR spectra of ethanol (blue), hexane (red) and acetone (green) extracts for Kompolti variety.
Figure S3 Comparison between the enlarged region [6.25 ppm - 6.5ppm] of the 1H NMR spectra from hexane extract of Tiborszallasi (blue) and Kompolti (purple) variety. A broad peak isolated at 6.40 ppm corresponding to the proton H-10 of Δ9-THC appears in the proton spectra of Tiborszallasi while this signal was undetectable in the 1H NMR spectrum acquired for Kompolti.
- I would suggest to use the cannabinoid numbering systems used in the recent review Prod. Rep., 2016, 33, 1357–1392.
Authors: We followed the reviewer's suggestion and we changed the numbering in fugure1, in table 2 e in the text in red. The new figure 1 and the new table 2 are:
Figure 1. Chemical structure and nuclei numbering of molecular fragments in hemp principal cannabinoids.
Table 2. 1H and 13C chemical shifts of the main cannabinoids in flowers extracts of Cannabis sativa (Tiborszallasi, variety) in CDCl3
|
Compound |
δ 1H ppm (multiplicity*, 1H-1H J-Coupling - Hz) |
δ 13C ppm |
||
|
CBD |
H3 |
3.86 (ddt; JH3-H4 =13.00 Hz (d), JH3-H2 = 3.51 Hz (d), JH3-H5 = 2.51 Hz (t)) |
C3 |
37.01 |
|
H2 |
5.57 |
C2 |
124.14 |
|
|
H6a |
2.05 -2.09 |
C6 |
30.36 |
|
|
H6b |
2.22 |
|||
|
H5 |
1.78-1.84 (ddd; JH5-H4 = 5.30 Hz (d), JH5-H6a =1.30Hz (d), JH5-H6b = 0.60Hz (d)) |
C5 |
28.35 |
|
|
H4 |
2.40 (dd; JH4-H3 =13.00Hz (d), JH4-H5 = 5.00 Hz (d)) |
C4 |
46.16 |
|
|
H7 |
1.79 (d ;3JH7-H2 0.50 Hz) |
C7 |
23.69 |
|
|
H9trans |
4.64 (dq ; J9trans-9cis = 2.65 Hz (d), 3J9trans-10 =1.50 Hz (q)) |
C9 |
110.81 |
|
|
H9cis |
4.53 (dq ; J9cis-9trans = 2.65 Hz (d), 3J9cis-10 = 0.92 Hz (q)) |
|||
|
H10 |
1.66 (dd ; 3J10-9cis = 0.92 Hz (d), 3J10-9trans =1.50 Hz (d)) |
C10 |
20.30 |
|
|
H2’ |
6.26 |
C2’ |
109.56 |
|
|
H4’ |
6.16 |
C4’ |
107.92 |
|
|
H1” |
2.43 (t) |
C1” |
35.46 |
|
|
H2” |
1.52-1.61 |
C2” |
30.65 |
|
|
H3”, H4” |
1.27-1.32 |
C3” |
31.48 |
|
|
C4” |
22.54 |
|||
|
H5” |
0.86-0.88 |
C5” |
14.04 |
|
|
CBDA |
H3 |
4.08 |
C3 |
35.38 |
|
H2 |
5.55 |
C2 |
124.14 |
|
|
H6a |
2.05-2.09 |
C6 |
30.36 |
|
|
H6b |
2.22 |
|||
|
H5 |
1.79 (ddd; JH5-H4 = 5.30 Hz (d), JH5-H6a = 1.30Hz (d); JH5-H6b = 0.60Hz (d)) |
C5 |
28.35 |
|
|
H4 |
2.40 (dd; JH4-H3 =13.00Hz (d), JH4-H5 = 5.00 Hz (d)) |
C4 |
46.45 |
|
|
H7 |
1.79 (d ; 3JH7-H2 = 0.50 Hz) |
C7 |
23.69 |
|
|
H9trans |
4.51 (dq; 3J9cis-9trans = 3.00 Hz (d) ; 3J9trans-10 =1.76 Hz (q)) |
C9 |
111.21-111.25 |
|
|
H9cis |
4.39 (dm ; 3J9cis-9trans 3.00 Hz (d)) |
|||
|
H10 |
1.70 |
C10 |
18.91 |
|
|
H4’ |
6.21 |
C4’ |
111.21-111.25 |
|
|
H1”a |
2.81 |
C1” |
36.68 |
|
|
H1”b |
2.92 |
|||
|
H2” |
1.52- 1.61 |
C2” |
31.24 |
|
|
H3”, H4” |
1.27-1.32 |
C3” |
31.94 |
|
|
C4” |
22.54 |
|||
|
H5” |
0.86-0.88 |
C5” |
14.04 |
|
|
CBG |
H2 |
6.24 |
C2 |
108.25 |
|
H5’, H4’ |
2.04 |
C4’ |
32.28 |
|
|
C5’ |
26.51 |
|||
|
H6’ |
5.12 |
C6’ |
125.08 |
|
|
H8’, H10’ |
1.68 |
C8’ |
20.51 |
|
|
C10’ |
23.44 |
|||
* Abbreviations: d, doublet; t, triplet; q, quadruplet; m, multiplet; dd, doublet of doublet; ddd, doublet of doublet of doublet; ddt, doublet of doublet of triplet; dq, doublet of quadruplet; dm, doublet of multiplet.
- In the NMR assignment discussion I would suggest to use the notation H-9 instead H9. Normally the subscript indicates the number of atoms.
Authors: We have modified in the text all the notation of the atoms according the suggestion of the referee. All changes are in red.
The English is rough in this manuscript but it's suitable for review purposes. The language issues can be dealt with in review.
Authors: We have revised the English in the revised version of the manuscript.

Round 2
Reviewer 1 Report
Dear Author(s)
After an exhaustive revision, the manuscript is Accept in present form. The resubmitted manuscript has been completely improved compared to its previous version. Therefore, the manuscript can be published in “Molecules”.
Best regards

Reviewer 2 Report
The authors properly revised the manuscript entitled “NMR Spectroscopy Applied to Metabolic Analysis of Natural Extracts of Cannabis sativa” according to the revisions suggested. The revised version of the manuscript results improved and appropriate to be published on Molecules journal.